

# Evaluation of ECMWF IFS-AER (CAMS) operational forecasts during cycle 41r1 - 46r1 with calibrated ceilometer profiles over Germany

Harald Flentje[1], Ina Mattis[1], Zak Kipling[2], Samuel Remy[3], and Werner Thomas[1]

[1]Deutscher Wetterdienst, Met. Obs. Hohenpeißenberg, D-82383 Hohenpeißenberg, Albin-Schwaiger-Weg 10, Germany
[2]European Centre for Medium-Range Weather Forecasts, Shinfield Park, Reading RG2 9AX, United Kingdom
[3] (HYGEOS (, 165 Avenue de Bretagne, Lille, France

*Correspondence to:* Harald Flentje (harald.flentje@dwd.de)

**Abstract.** Aerosol forecasts by the European Center for Medium Range Weather Forecasts (ECMWF) Integrated Forecasting System IFS-AER for years 2016-2019 (cycle 41r1 - 46r1) are compared to vertical profiles of particle backscatter from the Deutscher Wetterdienst (DWD) ceilometer network. The system has been developed in the Copernicus Atmosphere Monitoring Service (CAMS) and its

precursors. The focus of this article is to evaluate the realism of the vertical aerosol distribution from 0.3 to 8 km above ground, coded in the shape, bias and temporal variation of the profiles. The common physical quantity, the attenuated backscatter $\beta^*(z)$ , is directly measured and calculated from the model mass mixing ratios of the different aerosol types using the model's inherent aerosol microphysical properties.

Pearson correlation coefficients of daily average simulated and observed vertical profiles between r=0.6-0.8 in summer and 0.7-0.95 in winter indicate that most of the vertical structure is captured. It is governed by larger $\beta^*(z)$ in the mixing-layer and comparably well captured with the successive model versions. The aerosol load tends to be high-biased near the surface, be underestimated in the mixing layer and realistic at small background values in the undisturbed free troposphere. A seasonal

cycle of the bias below 1 km height indicates that aerosol sources and/or lifetimes are overestimated in summer and pollution episodes not fully resolved in winter. Long-range transport of Saharan dust or fire smoke is captured and timely, only the dispersion to smaller scales is not resolved in detail. Over Germany $\beta^*(z)$ from Saharan dust and sea salt are considerably overestimated. Differences between model and ceilometer profiles are investigated using observed in-situ mass concentrations

of organic, black carbon, $SO_4$, $NO_3$, $NH_4$ and proxys for mineral dust and sea-salt near the surface. Accordingly, $SO_4$ and OM sources as well as gas-to-particle partitioning of the $NO_3$-$NH_4$-system



are too strong. The top of the mixing layer on average appears too smooth and few 100 m too low in the model. Finally, a discussion is included of the considerable uncertainties in the observations, the conversion from modeled to observed physical quantities, and from necessary adaptions of varying resolutions and definitions.

## 1 Introduction

Aerosol particles play a key role in atmospheric processes depending on scales, heights and geographical regions. They affect climate and weather, directly by light scattering and absorption (Hansen et al., 1997; Ramanathan et al., 2007; WMO, 2013), indirectly by altering formation and droplet size of clouds (Lohmann et al., 2007), and by their impact on saturation and vertical exchange (Ackerman et al., 2000). In the lower troposphere particle emission, transport and heterogeneous chemical processes degrade air-quality and health (Galanter et al., 2000; Karanasiou et al., 2012; Pèrez et al., 2012), but then particles mediate gas-to-particle conversion, scavenging and final removal of trace gases from the atmosphere (e.g. Birmili et al. (2003); Kolb and et al (2010)). Their largely variable primary (direct) and secondary (precursor-initiated) sources, both natural and anthropogenic, resulted in a bunch of measurement- and numerical modeling techniques to analyze and understand particle atmospheric abundance and effects. Actually, increasing emissions of anthropogenic gases and aerosols during the last century has made aerosols a key trigger of severe pollution episodes and one of the largest uncertainties in assessments of climate change (e.g. Gilge et al. (2010); WMO (2013)).

Largest primary natural sources of atmospheric particles are oceans (sea salt), arid regions (mineral dust), boreal forests (organic and biomass-burning) and volcanic eruptions. Except volcanoes, these depend on season and weather, and all are linked to specific, partly extended source regions. Anthropogenic primary sources also exhibit regular spatial and temporal patterns, while secondary production is mainly driven by dispersion of precursors/condensables, transport and radiation. Globally, wet sea-salt is the most abundant aerosol, but referred to the dry state mineral dust comprises an even larger portion of the global aerosol load (Kinne and et al, 2006; Huneeus and et al, 2011) and about one third of the primarily emitted aerosol mass (Houghton et al., 2001; WMO, 2013). Europe is reached by Saharan dust oftentimes per year (Moulin et al., 1998; Gobbi et al., 2000; Ansmann and et al, 2003; Collaud-Coen et al., 2004; Papayannis et al., 2008; Pey et al., 2013; Flentje et al., 2015) where, decreasing towards the north, it contributes between 5%-30% to the total dry particle mass (Putaud et al., 2010). Dust may carry bacteria and has been associated to dispersion of meningitis (Griffin, 2007). Open fire emissions are in ∼85% (globally) linked to land-use and agriculture in the tropics (Andreae and Merlet, 2001), but drought-induced wildfires and boreal burns (Damoah et al., 2004; Hyer et al., 2007; Stohl et al., 2002) are more effective in spring and summer for mid-latitudes (Trickl et al., 2015). Owing to their highly variable source characteristics and dispersion (Labonne





et al., 2007), fires constitute a challenge for climate- and aerosol/chemistry-transport-models (Generoso et al., 2008; Kaiser et al., 2012). The small sized BB particles can enter deep into lungs and plant stomata. Their fractal surfaces favor adsorption of harmful combustion by-products, lead-

ing to coatings that may cause respiratory, allergic, cardiovascular and cancerous diseases (Mölter et al., 2014). Along with this, air-quality regulations like the European Directive 2008/50/EG for $PM_{10}/PM_{2.5}$, are currently revised to tackle issues related to carbonaceous fine ($PM_1$) and ultrafine ($<0.1\mu$m ) particles (Linares et al., 2009). Extensive modeling efforts have been made to investigate the abundance of aerosols and their role in weather, climate and air-quality (Stier et al., 2005; Grell

et al., 2011; Wang et al., 2011; Zhang et al., 2012).

Numerical models associate emissions, transport, transformations and sink mechanisms, whereby each particle type poses its individual challenges. Comprehensive views over capabilities of present numerical models to simulate radiative properties of particles and their meteorological relevance can be found in Morcrette et al. (2009) or Baklanov et al. (2014). Other studies discuss more specific

impacts of mineral dust e.g. Pèrez et al. (2006b), sea-salt (precursors) e.g. O'Dowd et al. (1997) and forest-fires on regional weather, (e.g. Andreae and Merlet (2001); Stohl et al. (2002); Andreae and Rosenfeld (2008)). In this context it is important to minimize uncertainties of simulated aerosol optical properties arising from assumptions on their physical and chemical composition (Curci et al., 2015). The latter typically stem from emission inventories (Granier and et al, 2011; EDGAR, 2013;

Gidden et al., 2019), implemented source functions (Dentener et al., 2006; Morcrette et al., 2009, 2011; Spracklen et al., 2011) or data assimilation (Benedetti et al., 2009; Kaiser et al., 2012). All these approaches are applied by the comprehensive forecasting system for regional and global scales that has been developed in the series of PROMOTE, GEMS, MACC I-III projects for the Copernicus Atmosphere Monitoring Service (CAMS - https://atmosphere.copernicus.eu/charts/cams/) at the Eu-

ropean Center for Medium Range Weather Forecast (ECMWF) (Morcrette et al., 2009; Flemming et al., 2017; Rémy et al., 2019). A discussion of discrepancies and synergies between inventory-based 'bottom-up' and observation-based 'top-down' estimations of anthropogenic emissions is given in (van der Gon et al., 2012). Owing to the rapid changes of emissions, the inventories in the CAMS system have considerably evolved since 2005 as documented on the ECMWF website

(https://confluence.ecmwf.int/display/COPSRV/CAMS+Global). The same holds for the representation and parameterisation of physical and chemical processes as well as temporal and spatial resolution of the model grid (Rémy et al., 2019). An important process that rapidly modifies particle properties (e.g. turbidity) is the water uptake/release driven by hygroscopic fractions (Weingartner et al., 2002; Swietlicki et al., 2008; Hong et al., 2014; Chan et al., 2018). Another one of public

attention is cloud formation during dust events, that regularly causes noticeable wrong forecasts. Often the spatial-temporal knowledge of sources, properties and dispersion of particles limits the accuracy of corresponding forecasts. As the benefit of 'chemical weather' forecasts for most users depends on timeliness (near-real time dissemination) and reliability, assessments of uncertainties in-





volved in atmospheric composition modeling (Ordonez et al., 2010) inclusive their meteorological

driver (e.g. Flentje et al. (2005, 2007)) with aid of independent observations is essential .

Global aerosol model evaluations so far concentrate on aerosol optical depth (AOD) measurements e.g. from AERONET (Holben and et al, 2001; Cesnulyte et al., 2014), however limited to daytime (except few moon-radiometers) and without resolving the vertical distribution. Regional models mostly think and verify in terms of particulate matter mass concentration $PM_{10}$ or $PM_{2.5}$, often

without resolving composition and sizes of particles (Stidworthy et al., 2018; Akritidis et al., 2018). Assessments of detailed particle properties typically suffer from low availability and representativeness of observations and are applied retrospectively (Flemming et al., 2017; Inness et al., 2019). Discrepancies between model and observations are typically reported in evaluation of operational or specific aerosol transport of aerosols (Pèrez et al., 2006a; Morcrette et al., 2011; Basart et al.,

2012). They result from displacements, time-shifts, misses or excess plumes/layers, since necessarily rough parametrisations of complex source, transformation and sink mechanisms sometimes cannot fully reproduce the highly variable dispersion of particles in the atmosphere. Simplified representations of soil conditions, mobilisation, emission, convective/turbulent lifting and diffusion affect the source strengths, shortcomings in dry and wet deposition as well as coagulation and chem-

ical transformations propagate to sinks and range (Tegen and Schepanski, 2009). Only recently, aerosol profiles have been considered, measured by lidars (GALION - WMO-GAW Report No. 178) and ceilometers (Benedetti et al., 2009; Wiegner and Geiß, 2012; Wiegner et al., 2014; Chan et al., 2018), whereby the former are operated spatially sparse and temporally intermittent, the latter have no independent capability to identify and quantify particles and both do at best capture part

of the surface layer. Nonetheless, combined lidar and ceilometer networks are optimally suited for localizing and tracking aerosol plumes throughout the troposphere and for assimilation of aerosol information into atmospheric models (Benedetti et al., 2009; Bocquet et al., 2015).

The vertical profile particularly holds information about long-range quasi horizontal- as well as vertical exchange, altitude dependent exposure to pollutants, the impact of optically active particles on

stratification and radiation, cloud formation potential, dispersion of hazardous particles like volcanic ash, and by all these often allows source attribution more precise than from surface network observation alone. Inherently, it reflects the pollution and mixing state as well as the height of the planetary boundary layer or mixing layer ML. The air-quality- and health-impact of aerosols concentrates in the ML, which is defined as the lowest atmospheric layer where turbulent mixing occurs (White

et al., 2009). The mixing layer height (MLH) determines dilution of surface emissions, and whether lofted layers are entrained and mixed to the ground. During daytime the ML tends to be labile stratified and well mixed overland as a result of solar heating and convection. At night a shallow stable layer forms near the surface due to radiative cooling, where mixing only occurs as intermittent, shear-driven turbulence, well separated from a neutrally stratified residual layer above (Stull, 1988).

The MLH is used by aerosol and chemistry transport models to constrain the vertical exchange and


to scale the dispersion of reactive gases and aerosols (Monks et al., 2009) as well as greenhouse gas concentration budgets (Gerbig et al., 2008). As far at it reflects in temporally consistent aerosol gradients, it can be inferred from lidar/ceilometer profiles (Münkel et al., 2007; Haeffelin et al., 2012).

In this study, we primarily use ceilometer profile data from the German ceilometer network to evaluate CAMS aerosol forecasts. Section 2 describes the data and the methods to compare particle information in the CAMS global model with respect to observations. The results are presented in section 3 together with some auxilliary data sets for interpretation and are discussed in the context of possible model improvement in chapter 4. Key findings are summarized with an outlook to up-
coming activities in chapter 5.

## 2  Data sets and methodology

### 2.1  The CAMS aerosol model

The CAMS aerosol model system IFS-AER is described in Benedetti et al. (2009); Morcrette et al.
(2009); Rémy et al. (2019). Further information as well as analyses, forecasts, evaluation results and other products can be found on the web page https://atmosphere.copernicus.eu/. We use the operational runs with assimilation (ASM) from 01/2016 (cycle 41r1) to 12/2019 (cycle 46r1) and corresponding unconstrained control runs (CTR) as listed in Table 1 and in Table 3 in Rémy et al. (2019). The data were re-sampled from the reduced Gaussian grid at T255 to
$1.0° \times 1.0°$ resolution before 06/2016 and T511 to $0.5° \times 0.5°$ thereafter. Conceptually, regional models build on the global forecasts and refine these scales to few km but yet provide only aggregated aerosol quantities $PM_{2.5}$ or $PM_{10}$ rather than direct backscatter output nor the information necessary for conversion. The global aerosol model uses 14 prognostic variables: (3 size bins each of dust and sea-salt, hydrophilic/hydrophobic black and organic carbon, sulphate $SO_4$,
and as of 9 July 2019 (cycle 46r1) also nitrate $NO_3$ and ammonium $NH_4$). MODIS AOD and since cycle 45r1 also the Polar-Multi-Angle Product (Popp, 2016) are assimilated, optionally by 4D-Var (Benedetti et al., 2009) or the 3-D fields from the previous forecast. Owing to an adverse effect on headline scores during tests with CALIOP-lidar backscatter-profiles (1D-Var), yet no aerosol profiles are assimilated (Benedetti et al., 2009). As described in detail by Granier
and et al (2011); EDGAR (2013); Rémy et al. (2019) and documented at the ECMWF website (https://confluence.ecmwf.int/display/COPSRV/CAMS+Global/) aerosol sources in IFS-AER continuously develop with emission inventories EDGAR, MACCity(+SOA), CAMS_GLOB_ANT/BIO vx.x (anthropogenic/biogenic), stem from scaled fire emissions of the Global Fire Assimilation System GFAS (Kaiser et al., 2012) or are for dust, sea-salt and biogenic particles calculated from the
meteorological fields and surface conditions. Volcanic emission can be activated on demand. Hori-





zontal and vertical transport is based on the dynamics of the ECMWF model, complemented by vertical diffusion/convection, sedimentation and dry/wet deposition by large-scale and convective precipitation. Significant upgrades impacting our results are the increase of horizontal resolution from T255 to T511 after 06/2016, the coupling of organic matter emission to CO emissions (Spracklen
et al., 2011) as of 02/2017, the increase of vertical resolution from 60 to 137 levels and addition of $NO_3$ and $NH_4$ as of cycle46r1 in 07/2019 (cf. Table 3 in Rémy et al. (2019)). Based on the 00 UT analysis, 3-hourly profiles at time steps +3, +6, +9,... are extracted from 5-day forecast runs, such that noticeable adaptations by the analysis/assimilation are possible at 03 UT each. Ceilometer and model profiles/MLH are based on altitude above ground and model geopotential height, respectively.
The vertical displacement between the low-resolved model orography and real terrain height is only relevant for steep stations sticking out far above the model surface level, while over flat terrain this is below 100 m.

**Table 1.** Specification of CAMS model runs. For changes by successive cycles c.f. https://atmosphere.copernicus.eu/node/326/ and specifically for cycle 46r1 https://atmosphere.copernicus.eu/node/472/ as described in Table 3 in Rémy et al. (2019). ASM is like CTR, but additionally uses 4D-Var assimilation.

| Period | IFS-Cycle | Horiz Resolution | Levels |
|---|---|---|---|
| 01/2016-05/2016 | 41r1 | T255 - 1.0°x 1.0° | 60 |
| 06/2016-01/2017 | 41r1 | T511 - 0.5°x 0.5° | 60 |
| 02/2017-09/2017 | 43r1 | T511 - 0.5°x 0.5° | 60 |
| 10/2017-05/2018 | 43r3 | T511 - 0.5°x 0.5° | 60 |
| 06/2018-06/2019 | 45r1 | T511 - 0.5°x 0.5° | 60 |
| 07/2019-12/2019 | 46r1 | T511 - 0.5°x 0.5° | 137 |

### 2.1.1 Attenuated backscatter from model mass mixing ratios

The model provides mass mixing ratios $m_{p,i}$ of 14 particle types as output, which for comparison must be converted to a common physical quantity that is measured by the ceilometers. As such the attenuated backscatter $\beta^*(z)$ according to Eq. (1) is chosen rather than the backscatter coefficient $\beta(z)$ because it is the primary measured variable without assumptions and the model contains all information to calculate it:

$$\beta^*(z) = \beta(z) \exp\left\{ -2 \int_0^z \sigma(z') \mathrm{d}z' \right\} \tag{1}$$

Here $\beta(z)$ and $\sigma(z)$ are the backscatter and extinction coefficients, respectively. The further procedure is described in detail by Chan et al. (2018) and outlined here for easier tracking of the most important steps. At first the mass mixing ratios of the particle types are converted to mass concen-





trations $c_{p,i}$ by multiplication with the air density $\varrho_{\text{air}}$ as shown in Eq. (2).

$$c_{p,i}(z) = \varrho_{\text{air}}(z)\, m_{p,i}(z) \qquad \text{for i = 1,2,...,14} \tag{2}$$

Then the particle extinction coefficient $\sigma_{p,i}$ and the particle backscatter coefficient $\beta_{p,i}$ of each particle type $i$ have been pre-calculated using appropriate particle size distributions $dN(r)/dr$ and humidity dependent particle refractive indices $n$ as applied in IFS-AER (Chan et al., 2018). For consistency with the current implementation of the aerosols in the IFS model Mie scattering the-

ory has been applied for all particles. Model mass concentrations are then converted to extinction coefficients by means of the specific (mass) extinction coefficient $\sigma_{e,i}^*$ (unit: m$^2$/g)

$$\sigma_{e,i}^* = \frac{\sigma_{p,i}}{c_{p,i}} \tag{3}$$

Eq. (3) is applied separately to each size and humidity bin of the humidity dependent and size segregated particle types. For convenience the lidar ratio $S_{p,i}$ is commonly used to calculate particle

backscatter coefficients from extinction coefficients.

$$S_{p,i}(z) = \frac{\sigma_{p,i}(z)}{\beta_{p,i}(z)} \tag{4}$$

With this definition, the extinction and backscatter coefficients of each particle type are determined from Eqs. (5, 6).

$$\sigma_{p,i} = c_{p,i}\, \sigma_{e,i}^* \tag{5}$$

$$\beta_{p,i} = c_{p,i} \left( \frac{\sigma_{e,i}^*}{S_{p,i}} \right) \tag{6}$$

The contribution from air molecules is calculated according to Rayleigh theory using the following approximation for the molecular extinction coefficient $\sigma_m$ (in km$^{-1}$):

$$\sigma_m(z,\lambda) = 8.022 \cdot 10^{-4} \varrho_{\text{air}}(z) \lambda^{-4.08}$$

with the air density given in kg/m$^3$ and the wavelength $\lambda$ in $\mu$m. The profile of $\varrho_{\text{air}}$ is taken

from the IFS. The molecular lidar ratio $S_m$ is known to be $S_m = \sigma_m/\beta_m \approx 8\pi/3$. To increase computational efficiency the pre-calculated values of $\sigma_{e,i}^*$, $S_{p,i}(z)$ as well as $\varrho_{\text{air}}$ are stored in a look-up archive as displayed in Tables A1, A2, A3 in the appendix. In order to calculate the total $\beta^*(z)$ according to Eq. (1) the contributions from all particle types are summed up to yield the (total) backscatter coefficient:

$$\beta = \beta_m + \sum_{i=1}^{14} \beta_{p,i}$$

Finally, the attenuation is applied to $\beta(z)$ to calculate $\beta^*(z)$:

$$\beta^*(z) = \beta(z)\, exp \int\limits_0^z \frac{-2\sigma}{z}$$





### 2.2 Ceilometer network

The German Meteorological Agency (DWD) operates a network of about 160 Lufft-CHM15k ceilometers (∼60 in Jan 2016, Figure 1) which provide operational profiles of aerosol attenuated backscatter $Pz^2$ (Flentje et al., 2010a,b), available as quicklooks at http://www.dwd.de/ceilomap/ and the European pendant https://ceilometer.e-profile.eu/. CHM15k use a diode-pumped Nd:YAG solid state laser emitting at 1064 nm and range up to max. 15 km above ground. Typically, incomplete overlap in the near-field and low signal-to-noise ratio SNR in the far-field limit the inferable profile range to 0.3 - 8 km altitude Heese et al. (2010). The ceilometers of the network are operationally calibrated using the TOPROF/E-PROFILE Rayleigh calibration routine provided by MeteoSwiss. The Rayleigh method (Barrett and Ben-Dov, 1967) is applicable under clear sky and stable aerosol conditions, whereby only nighttime data averaged over 1-3 hours are used to avoid disturbance by background light. Rayleigh scattering profiles are calculated from National Center for Environmental Prediction (NCEP) and the National Center for Atmospheric Research (NCAR) reanalysis data. Though the low sensitivity of the IR wavelength to small particles $<0.1\mu$m limits Rayleigh calibration capability, it offers large contrast (SNR) against molecular scattering to track larger particles. System stability and output power monitoring allows to track the lidar constant $C_L$ and transfer the calibrations to daytime profiles (Böckman et al., 2004; Heese et al., 2010; Wiegner and Geiß, 2012; Wiegner et al., 2014). Only stations with a sufficient density of successful calibrations are considered. Attenuated backscatter $\beta^*(z)$ as a function of altitude z is then calculated from the background corrected ceilometer signals P(z) with the calibration constant $C_L$

$$\beta^*(z) = \frac{Pz^2}{C_L} \qquad (7)$$

The $C_L$ values are first cleaned for outliers (<>1.5 x 25th-75th percentile of 30-day average), smoothed with a 30-day running mean and finally interpolated to hourly values to be used in Eq. (1). The typical uncertainty of an individual calibration is 15–20 %, while the actual error is smaller due to the temporal smoothing. The accuracy of the retrieved backscatter linearly depends on the accuracy of $C_L$. The monthly variation of $C_L$ is usually less than 5% and the annual variation is 10–15 %. Finally, cloud-free attenuated backscatter profiles are averaged within $\pm$ 1 h around the corresponding model times. Profiles with precipitation, low clouds or instrument operation flags are excluded from the evaluation as far as possible but still produce occasional artifacts. The most prominent feature in the backscatter profiles usually is the planetary boundary layer, here identified with the aerosol mixing layer ML. Up to three aerosol layer-top heights MLH, calculated by a wavelet algorithm (Teschke and Pönitz, 2010), are reported by the instruments (cf. next section). Often the uppermost may be identified with the MLH, however ambiguities in the MLH definition and the different algorithms for its determination remain large (Haeffelin et al., 2012).





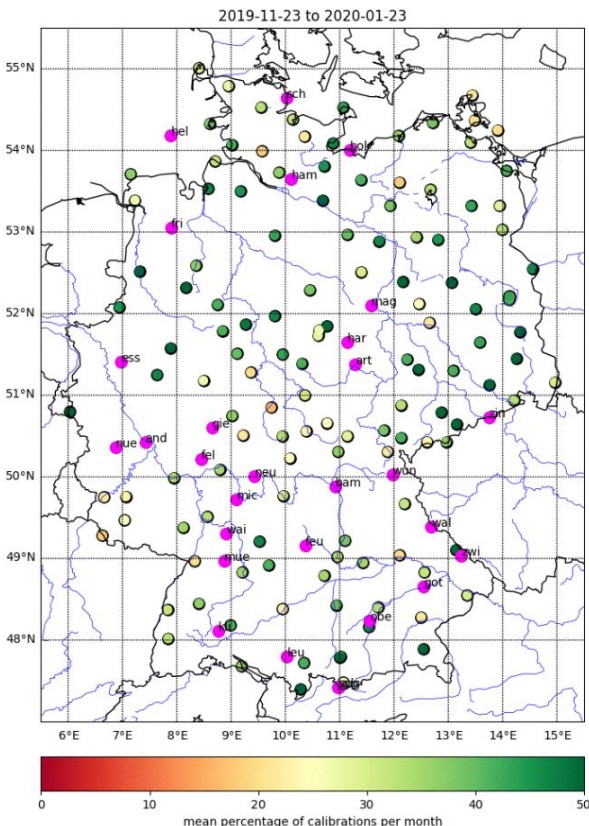

**Fig. 1.** Jenoptik CHM15k ceilometer network of the Deutscher Wetterdienst (DWD) in 2020, color coded by the number of available calibrations per month. Near real time quicklooks and metadata information are available via the website http://www.dwd.de/ceilomap/.

### 2.3 Comparison of mixing layer height MLH

The *model* mixing layer height (MLH) is extracted from the operational NWP forecast of the ECMWF Integrated Forecast System (IFS) model, archived at steps 3,6,9,...,24 h based on daily 00 UT anal-

ysis. The IFS model determines the MLH at the critical value $Ri = 0.25$ of the bulk Richardson number, which characterizes the degree of turbulence (Richardson et al., 2013). The vertical stability is estimated using the difference between each level and the lowest level. Several issues with this approach are described by e.g. Engeln and Teixeira (2013), related to the Richardson number being based on ratios of both dynamic and thermodynamic vertical gradients rather than of temperature

and/or humidity as such, the use of dry variables in cloudy situations, and the fact that the Richardson number as a measure of local turbulence is often unable to properly characterize the turbulent properties of convective boundary layers. Turbulent kinetic energy, which could better be used, how-





ever, is rarely used in global models and as such is not available (Engeln and Teixeira, 2013).

The MLH *observations* are based on two approaches: visual inspection of individual daily time-
height sections of $\beta^*(z)$ and data provided by the firmware implemented in the CHM15k ceilome-
ters. The former is elaborate, subjective and requires an experienced analyst of 2-D backscatter
sections. The latter is automated and precise but suffers from severe inaccuracies and ambiguities.
In principle, each MLH detection is a pattern recognition problem which is based on the assump-
tion that the vertical distribution of aerosol can be used as a tracer for boundaries. This, however,
is not always the case. The absolute value of the backscatter is typically not needed since the rel-
evant information seems to be completely coded in the gradient (but possibly of different orders)
of the backscatter profile (Teschke and Pönitz, 2010) and its temporal development. The CHM15k
firmware calculates MLH from the backscatter intensities of the ceilometer range corrected signal
($Pz^2$) by means of a wavelet transform algorithm (Teschke and Pönitz, 2010). Up to 3 layers with
quality flags are reported, which however lack specificity, temporal continuity and distinctness. In
this respect, Haeffelin et al. (2012) find in their analysis of limitations and capabilities of existing
mixing height retrieval techniques "'...no evidence that the first derivative, wavelet transform, and
two-dimensional derivative techniques result in different skills to detect one or multiple significant
aerosol gradients.'". The inferred MLH are mostly unrealistic in cases with multiple-layers, low
clouds/fog, small aerosol gradients, precipitation and long-range transport of e.g. dust, smoke etc.
Particularly Saharan dust days often must be excluded as the $\beta^*(z)$ gradient at the ML top disap-
pears. MLH below 400 m a.g. typically cannot be detected by the CHM15k because limited accuracy
of the overlap correction causes artifacts around that height. It turns out that the automatic data can
hardly be made a reliable reference, even when only subsets of clear cases and comparatively robust
metrics like maximum daily mixing layer heights MMLH are used. To this end visual inspection
of many individual cases illustrates why algorithms fail with ubiquitous complex scenes and at the
same time provides reasonable estimates of MMLH. Also owing to the sometimes thick entrainment
zone the uncertainty of visual MMLH detection ranges around $\pm$ 100 m for clearly developing ML.
Finally, it must be clearly stated, that the discussion of MMLH in this article is included as it is the
most prominent feature in the vertical profile, but it is not intended as a rigorous evaluation.

### 2.4 In-situ measurements of particle composition and -sizes

For comparison of modeled particle composition, in-situ particle measurements are used from the
German Global Atmosphere Watch (GAW) global station Hohenpeißenberg HPB (47.8°N, 11.0°E,
980 m a.s.l.), (Flentje et al., 2015). Hohenpeißenberg is a pre-Alpine hill, sticking out 300 m above
the surrounding forest/grassland and represents rural central European conditions. Measurements
of the particle composition (used for interpretation of the profile results) stem from Aerosol Chem-
ical Speciation Monitor Q-ACSM (Aerodyne Res. Inc.,Billerica, MA, USA; Ng et al. (2011)) and
quartz/teflon-filter probes analysed for water-soluble ions with a DIONEX ICS 1000 (Henning et al.,





2002), particle sizes from Optical Particle Sizer (OPS, GRIMM EDM180, Ainring, Germany) as de-
tailed in Flentje et al. (2015). These in-situ measurements' sensitivity and accuracy limitations are
small compared to other uncertainties of global model evaluation. Only the model altitude-level of
correspondence is not unambiguously to determine for mountain stations sticking out from the model
orography. The profile evaluation circumvents this by excluding stations in steep terrain and through
the negligible effect of the orography at higher altitudes. The compromise for HPB ($z_{obs} = 995$ m
a.s.l., $z_{geopot-model} = 665$ m) to capture both, surface effects and ambient conditions at elevated
sampling level, corresponds to L54-60 and L127-137 for the 60L and the 137L model version, re-
spectively, see e.g. Wagner et al. (2015). The range of concentrations within these altitudes indicates
the uncertainty.

### 2.5 Concept of evaluation

Given the complexity of spatio-temporal variations of 14 interacting aerosol types in the IFS-AER
model, it is important to breakdown the evaluation to a meaningful subset of metrics and scores and
adapt to the information content of the observation data. In this study, the evaluation focuses on the
vertical aerosol distribution and the altitude-dependence of the model-observation differences (bias)
from about 0.3 to 6 km above ground. Below 0.3 km, the incomplete overlap cannot be corrected
with sufficient accuracy. Above 6 km ceilometer data suffer increasingly from low signal/noise ratio
and cloud artifacts. To avoid sensitivity of our results to truncated profiles extending vertically over
less than 0.6 km or containing clouds and possibly falling precipitation streaks, such profiles are
excluded (cf. Section 4.2). In the vertical we distinguish the surface layer SL where the sources of
most particles are, the planetary boundary- or mixing-layer ML, and the free troposphere FT, where
long-range transport takes place. Model biases at certain altitudes can indicate specific deficiencies
in the model, but may also stem from uncertainties in the observation data, from inherent conver-
sions necessary to refer to the same physical quantity (mass mixing ratios versus scattering) or arise
from metrics and the methodology itself - this is discussed in Section 4.2.

While there are several options to discuss the agreement of forecast and observed backscatter pro-
files, we use the following metrics and scores to reveal different aspects of aerosol representation
by the IFS-AER model: The *correlation* of model-vs-observation profiles evaluates their shape, i.e.
efficiency and timeliness of vertical/horizontal transport, injection heights, representation of the mix-
ing layer and stratification. This can be jointly summarized in Taylor diagrams (Taylor, 2001) with
the standard deviation coding the variance/amplitude of the profiles. The bias or modified normal-
ized mean bias MNMB as a function of time and altitude evaluates the sources/sinks (-strength) and
physical and chemical transformations, separately for assimilation and control runs:

$$MNMB_{asm,ctr}(z,t) = 100 * \frac{2}{N} * \sum_{i=0}^{N} \frac{M_{asm,ctr}(z,t_i) - O(z,t_i)}{M_{asm,ctr}(z,t_i) + O(z,t_i)}$$

Either moving averages over certain altitude ranges (bias time series) or (e.g monthly) averages





resampled at the model levels (bias profiles) are calculated. The MNMB is used for comparability
within CAMS, because it is better suited to verify aerosol and chemical species concentrations com-
pared to verifying standard meteorological fields. Spatial or temporal variations can be much greater
and the model biases are frequently much larger in magnitude. Most importantly, typical concen-
trations vary quite widely between different aerosol types, regions and vertical levels, and a given
bias or error value can have a quite different significance. It is useful, therefore, to consider bias
and error metrics that are normalised with respect to observed concentrations and hence can provide
a consistent scale regardless of pollutant type, altitude or region (see e.g. Elguindi et al., 2010, or
Savage et al., 2013). Moreover, the MNMB is robust to outliers and converges to the normal bias for
biases approaching zero, while taking into account larger uncertainties in the observations and the
representativeness issue when comparing coarse-resolved global models versus site-specific station
observations.

Taylor polar plots combine two statistical measures for pairs of profiles, averaged over any optional
period of time (here daily means or medians) and a number of locations (stations with backscat-
ter profiles): the correlation of coincident pairs of modeled and observed vertical profiles plotted
along the azimuth, and the standard deviation of model profiles normalised to the observation on
the x-axis (Taylor, 2001). This means that correlation is calculated over altitude ranges rather than
periods of time. The ideal agreement or the reference point (observation) is thus located at polar
coordinate [1,1]. Noteworthy, the distance from the reference in Taylor polar plots corresponds to
the root-mean-square error RMSE. While Taylor plots powerfully display performance changes be-
tween model versions in a strongly aggregated way, temporal variation of the correlation is better
revealed by time series.

As interpretation of the time- and height-dependent bias requires knowledge of individual compo-
nents' behavior in terms of their mass concentrations, size distributions and periodicities, which is
not contained in the ceilometers' backscatter profiles, aerosol in-situ measurements at the Hohen-
peißenberg GAW global station are consulted. Associated limitations are discussed below. The
general approach in this article builds on the work of Chan et al. (2018) but allows to investigate
additional model details beyond those discussed in there and complements Flemming et al. (2017);
Rémy et al. (2019). These and our results are compared in the discussion.

## 3    Results

The consistency of IFS-AER forecasts with observations is evaluated using different related aspects:
the temporal development of $\beta^*(z)$ and its bias at selected levels, the vertical shapes of $\beta^*(z)$ -
profiles and their correlation, periodicity, specific particle events and the height of the ML. Then
we sort, link and aggregate these criteria for selected height regimes to extract clear measures of





performance. Longer temporal averages like seasonal cycle are disturbed by five model upgrades
within the 4 years period. By considering mean and median values, the skills with and without
(peaks of) events are distinguished, the latter representing more background conditions and less the
inter-annual variability of (mostly dust) events. Negative and positive biases are denoted as 'low-
bias' or 'high-bias', respectively, their absolute amount classified as large or small. The relative data
coverage of 3-hourly profiles from all stations remaining for evaluation is 93%, 92%, 89%, 83%,
71%, 46%, 16% at 0.4 km, 1 km, 2 km, 3 km, 4 km, 5 km, and 6 km above ground, respectively.

### 3.1   Bias and MNMB

Figure 2 shows the temporal evolution of bias (upper panels) and modified normalized mean bias
MNMB (lower panels), each for runs with assimilation (ASM - red/orange) and corresponding con-
trol runs (CTR - green/blue) around vertical model levels spaced by 1 km (0.4, 1, 2,...6 km above
ground), each averaged over $\pm$ 1 model level. The data averaged over 21 German ceilometer stations
becomes statistically sparse at higher levels $\geq$6 km. Analogous information but transformed to the
whole vertical profiles of monthly mean and median bias of $\beta^*(z)$ is shown in Figure 3 color coded
by months each for 2016 to 2019. The following results refer to both figures.



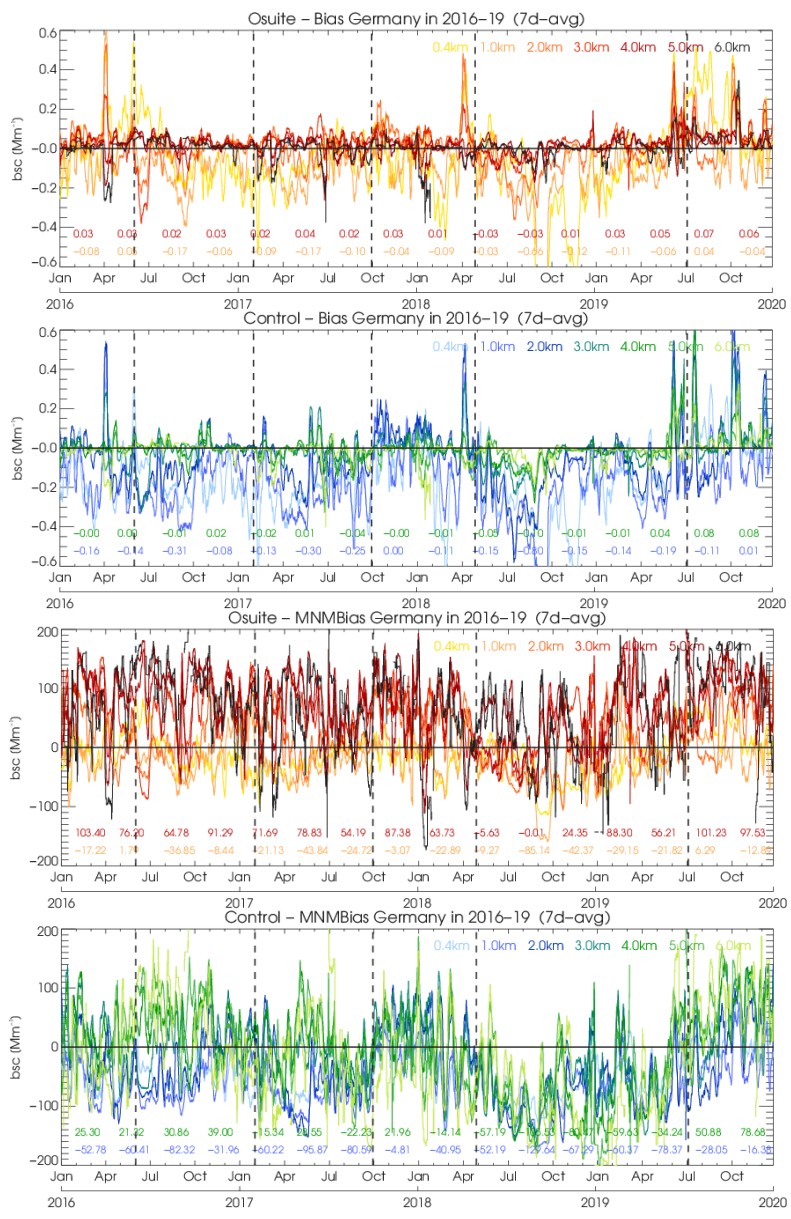

**Fig. 2.** 7-day running mean bias of $\beta^*(z)$ from ASM (1st panel) and CTR (2nd panel) combined from 21 German stations in 2016-2019. Same for modified normalized mean bias (MNMB) in 3rd and 4th panel. Colors refer to different altitudes above ground. Vertical black lines indicate major model updates as in Table 1.

Bias of $\beta^*(z)$ shows a clearly different behavior near the surface, in the ML and the FT, with
upward tendencies toward the surface, low bulges in the ML reaching up to ≈0.5-1 km in winter and





to ≈1-2 km in summer, and enhanced variation related to irregular long-range transport, mostly of dust, in the FT as shown in Figure 3. Estimated error bars overlayed to the CTR profiles indicate the significance of the biases. The low bias scatter above 6 km are artifacts caused by cloud boundaries not captured by the quality control. Owing to events, the mean bias is on average larger and scatters more than the median, particularly in the FT which holds little aerosol in undistorted situations. In several months Saharan dust events cause a high bias in the upper ML and the FT. A positive impact of the assimilation reflects in smaller and less variable bias in ASM than in CTR. This behavior is summarized in Tables 2 and 3 and clearly reflects in Figure 2 in the temporal development of bias and MNMB averaged over 7 days at different altitudes. One-day and 30-d moving average versions are shown for illustration in the appendix Section A. In Figures 2, A1a and A1b reddish colors refer to ASM and bluish colors to the control run without assimilation CTR. They show that both bias and MNMB tend to be lower in CTR than in ASM, particularly at lower heights, but also show a tremendous variability on daily time scales. Longer-lasting tendencies reveal only by averaging. Note in Figures 2 and 3 that only ASM is used with cycle 41r1 before June 2016. MNMB is less sensitive to vertical variation of the absolute values of $\beta^*(z)$ and is thus insensitive to the large vertical on-average decrease of the aerosol load. It shows for CTR a stronger vertical association after mid 2018 (Figure 2) and an overall downward trend 2016-2018 turning into an increase in 2019. In ASM this underestimation by trend and recovery is small and only evident in the FT. After the change to cycle 46r1 bias and MNMB in ASM and CTR approach each other at all altitudes.



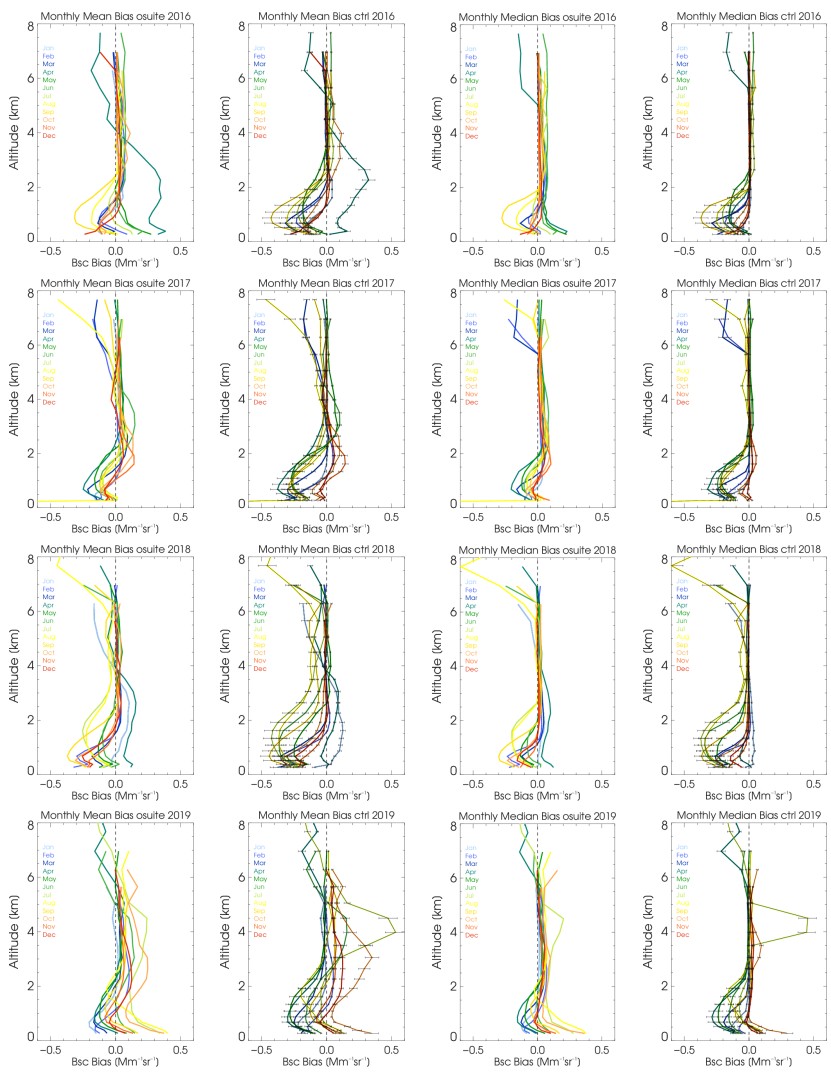

**Fig. 3.** Monthly mean (left pair) and median (right pair) profiles of bias ASM/ceilo (left) and CTR(geuh,...,h7c4)/ceilo (right), combined from 21 German stations in 2016-2019. At higher altitudes the profiles are partly contaminated by remaining cloud artifacts.

Over the four years monthly bias profiles have become more variable, the means more than medians and CTR more than ASM (Figure 3). This may reflect changes to model source strengths (OM and SD 02/17, $SO_4$ 10/17, sea-salt 06/18 - c.f. Table 3 in Rémy et al. (2019)), larger errors during more frequent events and a positive impact of the assimilation, respectively. This scatter is particularly observed in the ML where model $\beta^*(z)$ are on average lower than observed till 07/2019 and higher thereafter. Particularly CTR shows lower $\beta^*(z)$ bias and MNMB around summers at low





heights (MNMB around -100%), while ASM remains flatter thanks to the assimilation (Figure 2). The vertical gradient of MNMB is more stable on the long-term for ASM than for CTR. Surface

layer biases (SLB) stick out high (up to 0.3 Mm$^{-1}$sr$^{-1}$) with cycle 41r1 T255 in spring 2016 and with cycle 46r1 after 07/2019 (up to 0.4 Mm$^{-1}$sr$^{-1}$). In-between they were smaller or negative as shown in Figure 2 and Table 3. A clear bias increase with cycle 46r1, observed at 0.4/1 km, corresponds to overestimated $NO_3$, $NH_4$ and OM in the model as discussed with respect to GAW surface data in Section 3.3.


Though seasonal regularities are disturbed by five irregular model updates in the 2016-2019 period, bias/MNMB in ASM show opposing seasonal cycles in the lower (0.4 km a.g.) and the upper (2 km a.g.) ML with amplitudes of 0.2 Mm$^{-1}$sr$^{-1}$/40% (summer maximum) and 0.1 Mm$^{-1}$sr$^{-1}$/70% (summer minimum), respectively (Figure 4). Figure A1b shows that this is particularly clear before

cycle 43r3 in Oct 2017. The seasonal amplitude is small at the intermediate level 1 km a.g. The summer minimum reaches up to 3 km (MNMB even to 4 km a.g.), while it is variable dominated by Saharan dust events at 5 and 6 km a.g. Table 2 specifies summer and winter average biases range from -0.09 to 0.05 Mm$^{-1}$sr$^{-1}$ and -0.25 to 0.02 Mm$^{-1}$sr$^{-1}$ for ASM and CTR, respectively. A weekly cycle is neither significant in the bias nor MNMB, indicating a negligible influence of an-

thropogenic emissions on short-term variations. This is as expected because of the inventories' low temporal resolution (1 month).



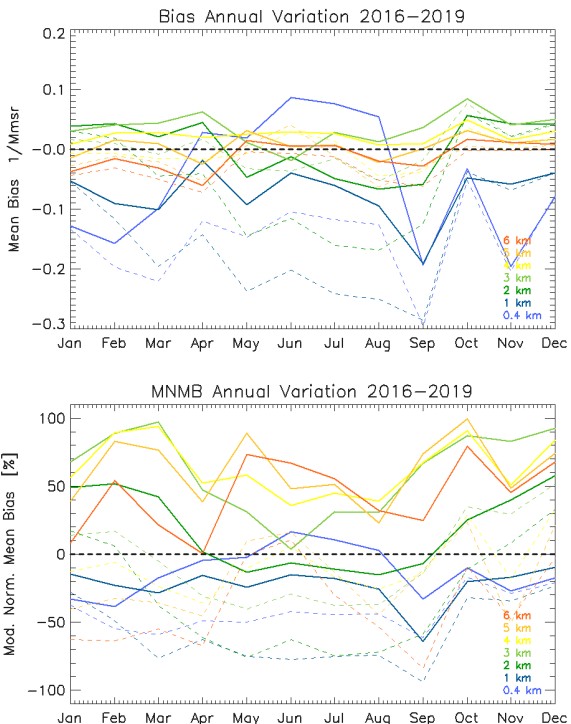

**Fig. 4.** Annual variation of bias and MNMB of $\beta^*(z)$ for ASM and CTR (dashed) combined from 21 German stations in 2016-2019.

The reported tremendous variability with season, altitude, events and model cycle is condensed to Tables 3 and 2 and summarized as follows:

– Near-Surface Layer ($\sim$0.4 km): bias decreases with height; high-bias with cycle 46r1 due to overestimated sources of $NO_3$, $NH_4$ and OM; bias seasonal cycle with summer-max and winter-min; pronounced low-bias in autumn 2018 (less in MNMB); mostly low-bias and low-MNMB except for summers 2016 and 2019 due to Saharan dust events; average MNMB range from -23% - 34% for ASM and from -68% - 16% for CTR

– Central Mixing Layer ($\sim$1 km): most pronounced low-bias in profile; no significant seasonal variation of bias/MNMB; average MNMB mostly negative between 1% to -33% for ASM and -18% to -82% for CTR; lowest bias in Q3 and Q4 of 2018

     – Entrainment Zone ($\sim$ 1 km in winter, $\sim$2 km in summer): transition from ML to FT coincides with bias drop to small values (except during SD events); As detailed in section 4.1 it's vertical

extension is too large





– Free Troposphere ($>3$ km): monthly median bias mostly not significant and $< 0.1\,\mathrm{Mm}^{-1}\mathrm{sr}^{-1}$; highly variable mean bias and MNMB due to SD events; average MNMB range from 29% - 99% for ASM and from -67% - 63% for CTR

Numerous periods with opposing high-bias in SL/ML and low-bias in FT indicate vertical displacement of layers within the profile. Quite expectable within individual profiles, it often also lasts for longer periods as can be see in the bias panels in Figures 2 as well as A1a and A1b, e.g. in Apr, May, Jun 2016 and repeatedly until cycle 45r1 in mid 2018 whereafter it seemly disappeared. In case of longer periods it is evident even in the monthly mean profiles in Figure 3. The effect may be attributed to adaptions by the assimilation of AOD which adds no height information. Temporal shifts between model and observation result in low/high oscillations (or vice versa) of bias but are mostly smoothed out on daily time scales. A possible quantification is discussed in Section 3.4.3 for Saharan dust events. Outstanding positive monthly bias profiles (cf. Figures 3 and 2) or peaks of positive bias related to Saharan dust events e.g. in Apr and Jun 2016, Jan and Apr 2018 and Sep 2019, are discussed in Section 3.4.1 below. Highly polluted air from eastern Europe, reaching Germany in Feb/Mar 2018 in the ML below 1.5 km height is not captured by the model, resulting in low-bias -0.3 $\mathrm{Mm}^{-1}\mathrm{sr}^{-1}$ in Feb/Mar profiles. As discussed in section 3.3 this air contained much ammonium and nitrate (as $NH_4NO_3$) in the ML, which had not been included in the model by that time. While emissions of organic matter (OM) were increased in Jan 2017 ($\approx$30-60% of aerosol mass), nitrate and ammonia were newly introduced in July 2019 ($\approx$10-30% of aerosol mass as $NH_4NO_3$ or $NH_4(SO_4)_2$) in the rural central European ML. ML biases correspondingly turned positive in July 2019 and converging ASM and CTR indicate that the correction by the assimilation reduced notably (Figure 2).

The ML is the most relevant part of the profile. Its depth, evident in the median bias profiles, is discussed in Section 4.1. Owing to the change of it's heights between winter and summer and it's limited accuracy, averages over the entrainment zone introduce quite arbitrary errors and are not interpreted. With cycle 46r1 after July 2019 $\beta^*(z)$ increased significantly in the SL and ML, but the large gradient over the very lowest model levels seems not realistic and indicates either deficiencies in strength or balance of sources and sinks and/or too slow vertical transport.





**Table 2.** Bias [$\text{Mm}^{-1}\text{sr}^{-1}$] and MNMB [%], each with standard deviation of $\beta^*(z)$ for ASM and CTR runs at 0.4, 1,..., 6 km altitude above ground, separately for summer (May-Aug) and winter (Nov-Feb) periods

| Alt | ASM-su | CTR-su | ASM-wi | CTR-wi |
|---|---|---|---|---|
| BIAS: | | | | |
| 0.4 km | $0.02 \pm 0.25$ | $-0.13 \pm 0.23$ | $-0.09 \pm 0.26$ | $-0.12 \pm 0.28$ |
| 1 km | $-0.09 \pm 0.20$ | $-0.23 \pm 0.20$ | $-0.05 \pm 0.19$ | $-0.07 \pm 0.21$ |
| 2 km | $-0.04 \pm 0.15$ | $-0.14 \pm 0.18$ | $-0.04 \pm 0.11$ | $0.02 \pm 0.13$ |
| 3 km | $0.01 \pm 0.13$ | $-0.04 \pm 0.15$ | $-0.04 \pm 0.07$ | $0.02 \pm 0.08$ |
| 4 km | $0.02 \pm 0.10$ | $-0.01 \pm 0.13$ | $-0.02 \pm 0.07$ | $0.00 \pm 0.07$ |
| 5 km | $0.00 \pm 0.11$ | $-0.01 \pm 0.13$ | $-0.01 \pm 0.07$ | $-0.01 \pm 0.07$ |
| 6 km | $0.00 \pm 0.08$ | $-0.03 \pm 0.09$ | $-0.01 \pm 0.08$ | $-0.02 \pm 0.08$ |
| MNMBias: | | | | |
| 0.4 km | $6.9 \pm 47.6$ | $-42.3 \pm 53.8$ | $-21.7 \pm 59.6$ | $-27.1 \pm 63.4$ |
| 1 km | $-18.6 \pm 41.7$ | $-71.8 \pm 52.4$ | $-10.4 \pm 74.6$ | $-26.2 \pm 75.0$ |
| 2 km | $-9.70 \pm 50.1$ | $-69.0 \pm 63.0$ | $57.3 \pm 74.9$ | $27.8 \pm 83.1$ |
| 3 km | $24.3 \pm 72.9$ | $-33.4 \pm 83.4$ | $82.7 \pm 77.3$ | $33.6 \pm 98.2$ |
| 4 km | $44.4 \pm 82.8$ | $-7.5 \pm 97.4$ | $73.2 \pm 91.0$ | $9.0 \pm 112$ |
| 5 km | $52.7 \pm 95.3$ | $-3.7 \pm 112$ | $65.0 \pm 95.3$ | $-12.9 \pm 114$ |
| 6 km | $56.4 \pm 98.7$ | $-15.8 \pm 117$ | $42.8 \pm 105$ | $-33.5 \pm 105$ |





**Table 3.** Upper part: Bias [Mm$^{-1}$sr$^{-1}$] and MNMB [%] of $\beta^*(z)$ for ASM and CTR runs at 0.4, 1 and 4 km altitude above ground averages within the different model configurations of Table 1. Lower part: Average correlation (cor), normalized standard deviation (nstd) and integrated bias of IFS-AER versus ceilometer $\beta^*(z)$ profiles for ASM and CTR runs over constant model configuration periods as defined in Table 1. The vertical range varies according to the individual situations, at maximum between 0.4 and 8 km altitude above ground.

|  | 41r1 (T255) | 41r1 (T511) | 43r1 | 43r3 | 45r1 | 46r1 |
|---|---|---|---|---|---|---|
| ASM bias |  |  |  |  |  |  |
| 0.4 km | 0.04 | -0.04 | -0.07 | -0.04 | -0.11 | 0.2 |
| 1 km | -0.01 | -0.08 | -0.11 | -0.01 | -0.12 | 0.02 |
| 4 km | 0.03 | 0.03 | 0.03 | 0.02 | 0.0 | 0.06 |
| CTR bias |  |  |  |  |  |  |
| 0.4 km | - | -0.16 | -0.21 | -0.07 | -0.21 | 0.09 |
| 1 km | - | -0.17 | -0.22 | -0.03 | -0.23 | -0.06 |
| 4 km | - | 0.01 | -0.02 | -0.01 | -0-03 | 0.07 |
| ASM MNMB |  |  |  |  |  |  |
| 0.4 km | 5 | -10 | -20 | -8 | -23 | 34 |
| 1 km | -6 | -15 | -30 | -4 | -33 | 1 |
| 4 km | 86 | 82 | 67 | 65 | 29 | 99 |
| CTR MNMB |  |  |  |  |  |  |
| 0.4 km | - | -47 | -68 | -20 | -54 | 16 |
| 1 km | - | -57 | -82 | -18 | -78 | -20 |
| 4 km | - | 34 | -2 | -6 | -67 | 63 |
| ASM cor | 0.76 | 0.74 | 0.65 | 0.79 | 0.73 | 0.71 |
| CTR cor | 0.77 | 0.75 | 0.65 | 0.79 | 0.73 | 0.72 |
| ASM nsd | 0.98 | 0.53 | 0.46 | 0.68 | 0.52 | 0.96 |
| CTR nsd | 0.95 | 0.51 | 0.45 | 0.79 | 0.53 | 0.99 |
| ASM ibias | 0.09 | -0.10 | -0.19 | -0.03 | -0.26 | 0.08 |
| CTR ibias | 0.08 | -0.10 | -0.19 | -0.03 | -0.26 | 0.08 |

Large high-biases are mostly linked to Saharan dust events (e.g. Apr & Jun 2016, Jun, Jul & Oct 2017, Jan & Apr 2018 and Jun/Jul & Oct-Dec 2019). Occasionally, SD particles induce cloud formation (e.g. 16/17 Oct 2017) which largely increases the $\beta^*(z)$ signal in spite of constant dust aerosol load - cf. discussion in Section 3.4.2. As there is no clear $\beta^*(z)$ threshold above which ceilometer data are filtered out and are ignored, such events produce an oscillating high/low-bias signal. Larger underestimation or low-biases, are found especially in the ML (1-2 km lines in Figure 2) over periods of several months (e.g. Aug/Sep 2016, Apr/May 2017, Jul-Oct & Dec 2018, Jan & Mar-May 2019). Pollution episodes with high PM advection from eastern Europe (e.g. Jan 2017, Jan-Mar 2018) are





not captured and reflect in short bias/MNMB dips (cf. section 3.3). Though questionable or cloudy
ceilometer data are largely blacklisted, still cloud artifacts combined with sparse data availability
cause sharp low-bias dips at higher altitudes $\geq$5 km in Figure 2.

## 3.2 Profile shape - Correlation

The Pearson correlation coefficient r of model-observation $\beta^*(z)$ profile pairs specifically quantifies
the covariance of vertical variability, i.e. the shape of the profiles, independent of the bias. The ML
and eventual particle plumes in the FT govern this correlation. Again, elimination of clouds and
the overlap range is essential. Apart large event-driven situational variability, the profile correla-
tion exhibits no long-term tendency but a clear seasonal cycle with better agreement in winter and
less in summer as shown in Figure 5. Overlayed in Figure 5 are vertical lines indicating season-
ally irregular model upgrades and mean values over the IFS cycle periods from Table 1. The mean
correlations within the IFS configuration periods r $=$ 0.76/0.74/0.65/0.79/0.73/0.71 vary not signif-
icantly, and their differences reflect the seasonal cycle rather than indicating changes of the model
performance. Individual (3-hly) profile pairs or longer temporal averages are considered, whereby
the former penalizes already small time-shifts or displacements (and yields lower r), and diurnal or
longer averages neglect influences from early/lagged transport as well as the dominant diurnal cycle
of the ML and are more sensitive to irregular events. On a monthly basis also median profiles are
considered to evaluate specifically the model background profile. These are shown for illustration of
the actual profile shapes in the appendix in Figures A2 - A5.



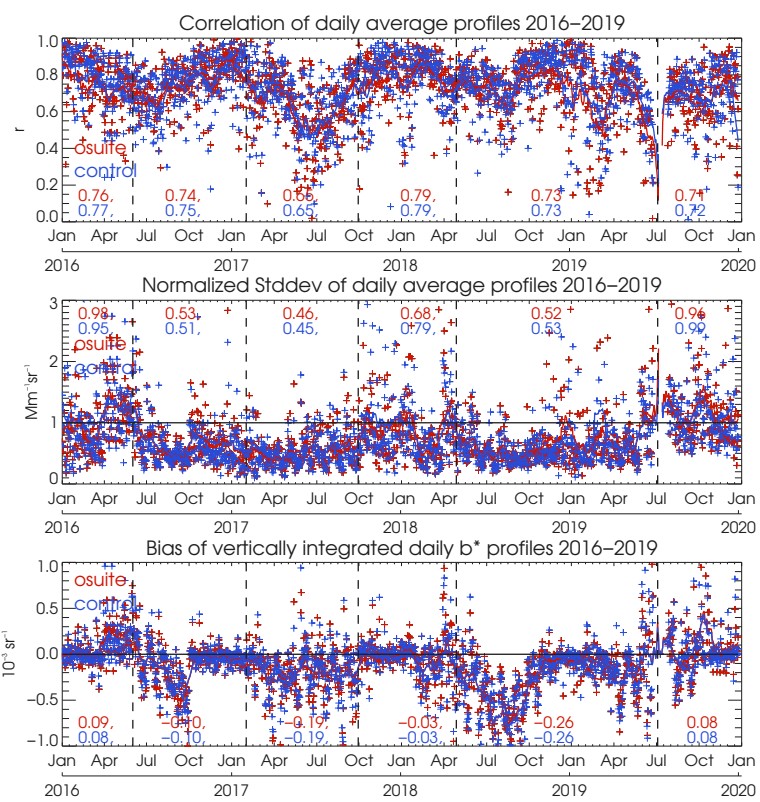

**Fig. 5.** Pearsons correlation coefficients (r - upper panel), standard deviation normalized towards ceilometer observations (mid-panel), and integrated bias of daily average $\beta^*(z)$ profiles of IFS-AER versus ceilometers for 2016-2019. Red crosses denote ASM, blue crosses the control run. The 3-day moving average line and median values over the periods with constant model configurations are added.

Generally, increasing correlation is found between IFS-AER fields and individual station profiles,

with longer averaging times: while only 50-60% of the observed 3-hourly vertical variability is explained by IFS-AER ($r_{3hly}$ =0.5 - 0.6), the explained fraction increases to 70-80% for diurnal average profiles ($r_{1dly}$ =0.7 - 0.8) as shown in Figure 5. Thus spatio-temporal aggregation defines the information to be revealed. Aerosol changes are very often not timely and/or (vertically) displaced on a few-hourly time scale, but longer (or more extended) events and developments are quite reliably

captured by IFS-AER. This is particularly true for Saharan dust transport where nearly all events are reproduced but the large concentrations - or $\beta^*(z)$ - combined with small scale inhomogeneity give rise to larger uncertainties as well (cf. Section 3.4.1). The mid panel of Figure 5 shows the variance





of daily average vertical profiles normalized to that of the observations as normalized standard deviation NSD. The time series and Table 4 reveals marked differences between the IFS cycles, given for ASM/CTR, separately: profile variance approaches the observations (NSD=0.97/0.93) during cycle 41r1 before June 2016 and NSD=0.95/0.96 during cycle46r1. Only about half the observed variance is simulated during cycles 41r1 after July 2016 (NSD=0.52/0.50), 43r1 (NSD=0.46/0.45), and 45r1 (NSD=0.51/0.52). Intermediate values (NSD=0.67/0.78) are found during cycle43r3. A similar measure like NSD (analog to AOD bias) is the vertically integrated $\beta^*(z)$ bias. It is dominated by the ML and/or events as in Figure 2 but has the limitation that every single profile has weather dependent vertical extension. No clear ruptures as for NSD appear at the model upgrade times for the integrated $\beta^*(z)$ diurnal profile bias in the 3rd panel of Figure 5 and extracted to Table 4. It is not clear whether this can be interpreted in terms of model upgrades where several adaptions of sources took place. For example sea salt as a large contributor to high $\beta^*(z)$ bias in the ML (Chan et al., 2018) was reduced inland after 06/2018 by re-distributing mass from fine to coarse particles (Rémy et al., 2019). As of 07/2019 $NO_3$ and $NH_4$ were added and probably too much as discussed in Section 3.3. On the other hand, the substantial increase of OM load in 02/2017, clearly evident at the surface (Section 3.3) seemly did not affect the profile integral.

A more condensed way than Figure 5 to descriptively visualize performance changes between model versions are Taylor polar plots as displayed in Figure 6 and explained in Section 2.5. Here, the average performance during the six IFS-AER configurations in Table 1 are summarized in terms of correlation, normalized standard deviation and the plotting-distance towards the reference, i.e. the root mean square error (Taylor, 2001). Accordingly, the model system has not systematically evolved towards improved representation of the profile shape, though mean values around $r_{1dly} = 0.7$ are already quite good. However, after some variation, finally the overall variance of the profile became nearly realistic on average after the implementation of $NO_3$ and $NH_4$ and adaptions to $SO_4$, organics and dust in cycle 46r1 in July 2019. The differences between ASM and CTR are small. It should be noted, that individual covariances of modeled and observed profiles vary quite strongly with time and location/station, meaning that many situations cannot be closely captured and even the observations may partly not be representative due to undetected artifacts (clouds, overlap correction, misalignment etc, not removed by the quality control).



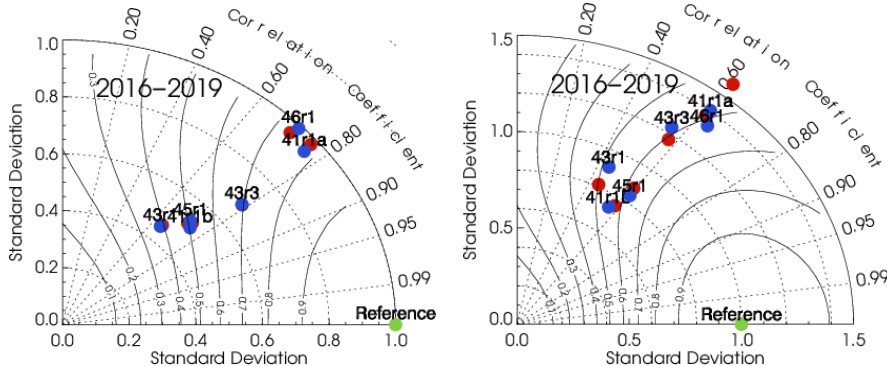

**Fig. 6.** Taylor plot combining Pearsons correlation coefficients (azimuth) and standard deviation normalized towards ceilometer observations (radius) from daily average $\beta^*(z)$ -profiles of IFS-AER versus ceilometers for 2016-2019. Left: median of all data, right: mean over 221 Saharan dust days as defined in Section 3.4.1. Red dots denote ASM, blue dot the corresponding control runs. Note the different x/y-axes.

### 3.3 Particle composition and -size at surface level

To better understand the differences between modeled and observed backscatter $\beta^*(z)$ profiles, near-surface mass concentrations MC of the prognostic aerosols in IFS-AER, namely $PM_{10}$, sulfate $SO_4$, nitrate $NO_3$, ammonium $NH_4$, black carbon BC, organics OM as well as qualitative proxys for 'sea-salt' SS and 'mineral dust' MD are compared to surface in-situ observations. All particle concentrations are modeled in IFS-AER and measured in situ in dry state without hygroscopic water

uptake. $PM_{10}$ is calculated from the model mass mixing ratios mmr according to the formula used in IFS-AER (Rémy et al., 2019): $PM_{10} = \rho([SS_1]/4.3 + [SS_2]/4.3 + [MD_1] + [MD_2] + 0.4[MD_3] + [OM] + [BC] + [SO4] + [NO3_1] + [NO3_2] + [NH4])$, whereby $[X_i]$ denotes the mmr of the i-th size bin of the size-resolved species and $\rho$ is the density of air. The 'dust' variable is not directly measured, but approximated by $MD = PM_{10} - [OM] - [BC] - [NO3] - [NH4] - [SO4] - [Cl]$ and

inferred on event basis to discuss contingency of events in Section 3.4.1. Mineral dust sizes at HPB are mostly smaller than 10 $\mu$m and its composition is largely disjunct from the other IFS-AER particle types. Chlorine Cl is used as a proxy for NaCl in sea salt, stoichiometrically corrected for the sodium Na portion ($m_{Na}/m_{Cl} \approx 22/35$) and for $\approx$7% of additional minor components $SO_4$, Mg, Ca etc. A rigorous evaluation of composition-resolved MC is beyond the scope of this article, but a

sanity check with data from the GAW global station Hohenpeißenberg (HPB) provides insight into the representation of individual aerosol types. According to the European aerosol climatology of Putaud et al. (2010), HPB is representative for central European rural conditions.





**Table 4.** Concentrations [$\mu g/m^3$] of IFS-AER prognostic aerosols by ASM and CTR versus GAW in-situ measurements at Hohenpeißenberg station, averaged over constant model configuration periods as defined in table 1.

|  | 41r1-T255 | 41r1-T511 | 43r1 | 43r3 | 45r1 | 46r1 |
|---|---|---|---|---|---|---|
| ASM PM$_{10}$ | 11.61 | 6.91 | 9.40 | 11.30 | 10.71 | 13.76 |
| CTR PM$_{10}$ | 10.43 | 5.55 | 6.37 | 10.06 | 6.92 | 11.36 |
| GAW PM$_{10}$ | 7.74 | 8.33 | 7.90 | 8.20 | 8.37 | 5.81 |
| ASM om | 1.01 | 1.50 | 4.08 | 5.93 | 6.06 | 4.74 |
| CTR om | 0.94 | 0.90 | 2.18 | 4.17 | 3.46 | 2.87 |
| GAW om | 2.52 | 2.63 | 2.71 | 2.63 | 3.10 | 1.79 |
| ASM bc | 0.54 | 0.61 | 0.56 | 0.33 | 0.30 | 0.18 |
| CTR bc | 0.50 | 0.49 | 0.20 | 0.36 | 0.15 | 0.11 |
| GAW bc | 0.35 | 0.47 | 0.35 | 0.46 | 0.39 | 0.30 |
| ASM SO$_4$ | 5.60 | 3.02 | 1.80 | 1.03 | 1.97 | 1.04 |
| CTR SO$_4$ | 4.60 | 1.46 | 0.70 | 0.78 | 0.78 | 0.40 |
| GAW SO$_4$ | 0.81 | 0.72 | 0.82 | 0.86 | 1.00 | 0.51 |
| ASM NO$_3$ | - | - | - | - | 3.21 | 3.95 |
| CTR NO$_3$ | - | - | - | - | 3.63 | 3.85 |
| GAW NO$_3$ | 0.70 | 1.22 | 0.95 | 1.67 | 1.53 | 0.81 |
| ASM NH$_4$ | - | - | - | - | 0.72 | 0.88 |
| CTR NH$_4$ | - | - | - | - | 0.80 | 0.87 |
| GAW NH$_4$ | 0.47 | 0.62 | 0.60 | 0.92 | 0.89 | 0.45 |
| ASM ss | 2.38 | 1.35 | 2.14 | 2.52 | 1.22 | 1.24 |
| CTR ss | 2.25 | 1.32 | 2.16 | 2.60 | 1.13 | 1.20 |
| GAW ss | 0.17 | 0.13 | 0.14 | 0.19 | 0.14 | 0.13 |
| ASM du | 4.34 | 1.41 | 2.42 | 2.82 | 2.44 | 6.04 |
| CTR du | 4.31 | 2.48 | 2.90 | 3.90 | 2.71 | 6.84 |
| GAW du | 1.94 | 2.78 | 2.28 | 1.58 | 2.69 | 1.79 |



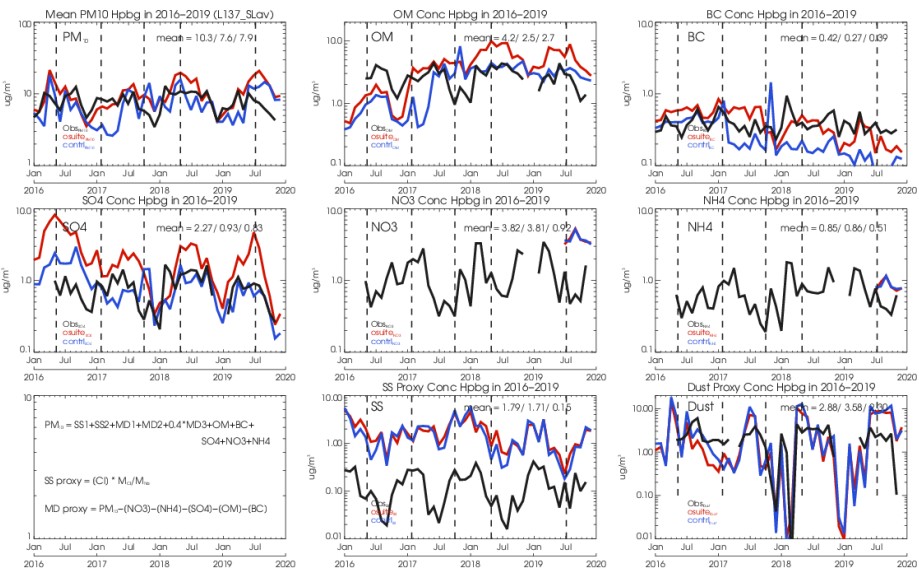

**Fig. 7.** Comparison of mass concentrations averaged over IFS-levels L54-L60/L127-L137 for L60/L137 model version and measured by ACSM and filter probes at GAW station Hohenpeißenberg for 2016 - 2019. From top/left to bottom/right: $PM_{10}$, OM, BC, $SO_4$, $NO_3$, and $NH_4$, chlorine, sea-salt- and dust-proxies as described in the text. Vertical black lines indicate major model updates as in Table 1. Note the different y-ranges.

As shown in Figure 7, the dry surface mass concentration $PM_{10}$ for ASM (10.3 $\mu g/m^3$) and CTR

(7.9 $\mu g/m^3$) roughly corresponds to HPB data (7.9 $\mu g/m^3$). The assimilation seems to bias surface concentrations a bit high. Species are detailed in Table 4. $PM_{10}$ approaches HPB data after the increase of OM with cycle 43r1 (02/2017), though this was partly compensated by a parallel decrease of $SO_4$, it is however overestimated as of cycle 46r1 after July 2019 due to introduction of $NO_3$ and $NH_4$, which are simulated roughly 3 $\mu g/m^3$ ($\sim 300\%$) and 0.3 $\mu g/m^3$ ($\sim 60\%$) too high at

HPB, respectively. Further changes with cycle 43r3 (10/2017) synchronize the phase but exaggerate the amplitude of the $SO_4$ annual cycle which together with the dominating high-biased contribution from OM causes most of the $PM_{10}$ overestimation near the surface in summers since 2018. After sulfate was reduced in cycle43r3 and beyond (Rémy et al., 2019), $SO_4$ in CTR agrees remarkably well with HPB while summer concentrations are by 2-4 $\mu g/m^3$ too high in ASM. BC which con-

tributes only about 5% in mass has evolved quite realistically with a slightly more decreasing trend in 2016-2019 than observed. Probably, emission inventories overestimate the decreasing trend over Europe where the decline has leveled off in the last decade.

Total suspended sea-salt is equally overestimated in ASM and CTR with mean MC around 1.8 $\mu g/m^3$, while the estimated abundance at the far inland HPS site is only 0.02-0.3 $\mu g/m^3$ with however



large error bars of $\pm$ 0.3 $\mu$g/m$^3$ due to the hard-to-sample coarse mode (5-20 $\mu$m) which contributes
about 0.3 $\mu$g/m$^3$ to the SS concentration in the model. The seasonal variation by roughly an order of
magnitude seems realistic. The large uncertainties and increases of bias in the PBL associated with
SS has already been discussed in Chan et al. (2018). To this, the above mentioned approximation
of SS via Cl has negligible impact. The observed-dust proxy contributes only 4-6% to the annual

average mass at HPB (Flentje et al., 2015). The seasonality is reproduced, but mean summer con-
tributions around 10 $\mu$g/m$^3$ would require much more events than observed and simulated, which
confirms that dust concentrations are overestimated not only near the surface but also in the higher
ML and the FT as noted in Section 3.1. The assimilation correction to dust MC of few $\mu$g/m$^3$ is too
small. These results are not affected by mass-to-backscatter conversion nor humidity and, due to av-

eraging over the lowest 300 m a.g., are not sensitive to the model level selected to represent surface
concentrations at HPB. The regional representativeness is limited to rural central Europe (Putaud
et al., 2010) where comparatively small concentrations prevail as discussed in Section 4.

### 3.4 Long-range transport

The DWD ceilometer network follows the 3-D dispersion of optically efficient particles like dust or

smoke and is therefore particularly suitable to verify the timelyness of long-range aerosol transport
in IFS-AER in a qualitative way. Against this, automated rendering of 2-D time-height sections from
the ensemble of stations to evolving 3-D fields is a challenge beyond the scope of this article, and
advanced metrics like fractions skill score (Roberts, 2008) have still to be adapted. Simpler options
to reduce the comparisons' degrees of freedom are to compare time-height slices at fixed locations

(stations), analyse representative cases or evaluate the representation of events qualitatively. In aged
air masses far from the sources, chemical transformations slow down and transport of particle lay-
ers/plumes becomes more passive. This reflects in wide consistency of aerosol fields in the IFS
model with large-scale dynamical structures in the middle and upper troposphere (e.g. Flentje et al.
(2005)).

### 3.4.1 Mineral dust

The previous sections showed that Saharan dust loads over Germany are over-estimated at the sur-
face and throughout the profile. The realistic seasonality (Figure 7) and the reasonable correlation
(Figure 5) however suggest that time and also vertical position of SD plumes are mostly captured
in IFS-AER, as long as the scales are sufficiently large. It can further be shown that IFS-AER fore-

casts have an extremely high score in capturing or reliably excluding significant Saharan dust days
SDD, which are inferred from the observations by visual inspection of 2-D network composite plots
(as in Figure 8) and backward trajectories and from the model by choosing a reasonable threshold
for maximum dust AOD within a box of 1° x 1° around selected ceilometer stations. Defining days
with max AOD$_{550nm,dust}$ > 0.03 and max AOD$_{550nm,dust}$ < 0.001 as SDD respective None-SDD





in the model and within the inherent uncertainties of type identification, these threshold yield 'ex-
cess' and 'miss' rates near zero, 221 'hit'-days and 271 zeroes. Hits and zeroes are SDD respective
clear days identified in both data sets, 'excess' SDD are simulated but not observed and 'misses'
denote observed SDD that are not reproduced by IFS-AER. Owing to the uncertain identification of
faint aerosol layers based on ceilometers + trajectories, the majority (2/3) of days in-between these
thresholds remain unclassified. This is however no severe limitation to this analysis, which is meant
to confirm qualitatively the high reliability of the forecasts w.r.t. decided SDD and non-SDD.

As several improvements were made to emission, size-distribution and (wet) deposition of dust
(Rémy et al., 2019), a Taylor diagram in Figure 6 for the subset of SDD with modeled maximum
$AOD_{550nm,dust} > 0.03$ shows the development of dust simulation by IFS-AER during the 2016-
2019 period. On SDD the correlation of profiles (shapes) is lower (r =0.4-0.6 instead of r =0.6-0.8)
while standard deviation (coding the amplitude of $\beta^*(z)$ ) is higher. The first indicates spatio-
temporal or vertical shifts of layers/plumes, the latter reflects overestimation of dust concentrations
but is not directly scaled to the SD bias due to the large influence of the ML on the profile. These
findings confirm the analysis by Rémy et al. (2019) who state good capability to reproduce dust
events as detected by AERONET station data (Holben and et al, 2001). According to the differ-
ent trajectories, the long-range transport pathway (via Atlantic, Mediterranean,...) does not effect
the accuracy of timing/positioning of plumes, while the scale reduction during regional stirring and
dispersion is the main reason degrading the representation of the vertical profile shape.

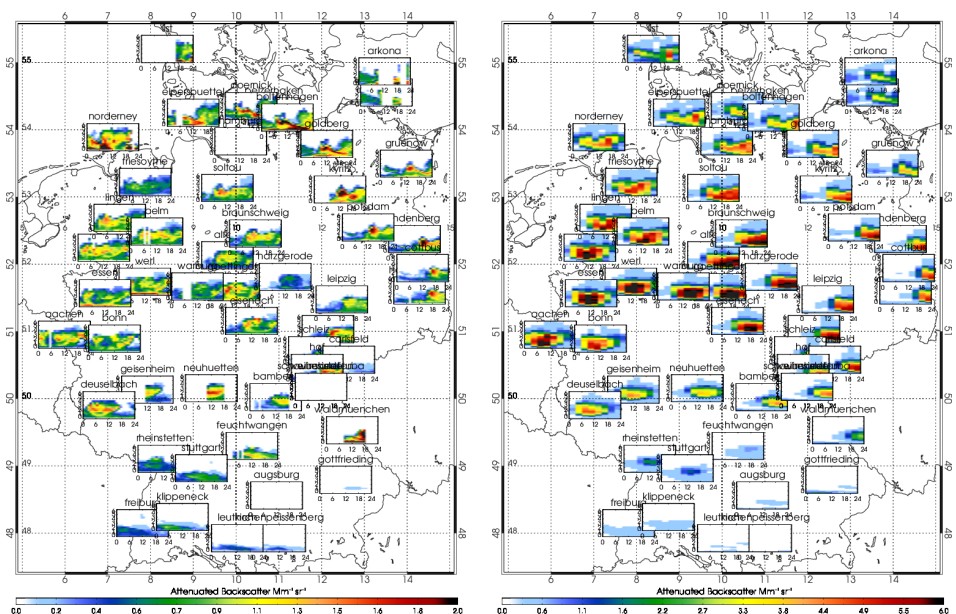

**Fig. 8.** Geo-referenced time-height plots of $\beta^*(z)$ at the stations of the German ceilometer network overview from 16 Oct 2017. Left: Ceilometer 2D-sections of $\beta^*(z)$ resampled to 3-hly resolution , right: corresponding IFS-AER 2D-sections.

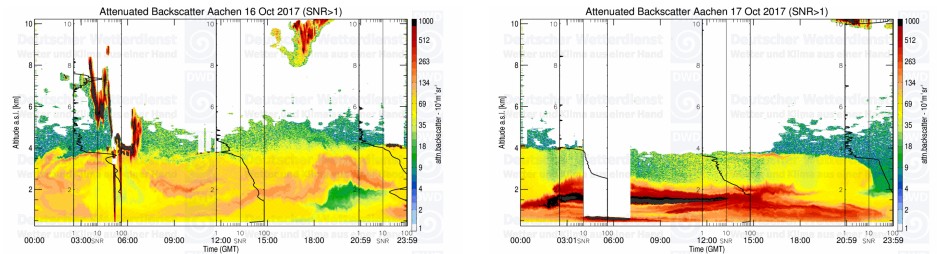

**Fig. 9.** Time-height sections of $\beta^*(z)$ at the ceilometer station near Aachen from 16 and 17 Oct 2017.

### 3.4.2 Cloud formation due to SD

Though Saharan dust transport is realistic in IFS-AER on spatio-temporal scales $>100$ km and $>1/2$ day, the dust load is mostly overestimated. Occasionally however, $\beta^*(z)$ of dust plumes is apparently underestimated because dust particles rapidly grow by water uptake and observed $\beta^*(z)$ changes though the dust mass concentration itself remains constant. Though the ability of (coated) mineral dust to foster cloud formation is well known, its simulation still is a challenge (Sassen et al., 2003;

Ansmann et al., 2005; Bangert et al., 2012). For example on 16 and 17 Oct 2017 a Sahara dust plume





swayed eastward over north-western Germany (Figure 8), shown in detail for the station Aachen in Figure 9. On both days similar dust loads, converted to similar $\beta^*(z)$ , are simulated by the model, but on 16 Oct observed $\beta^*(z)$ were as usual less than half of those modeled, while on 17 Oct hygroscopic growth or incipient cloud formation temporarily multiplied the optical signal tenfold

($\beta^*(z)\ _{max} = 1.2\ \cdot\ 10^{-5}Mm^{-1}sr^{-1}$) while the dust mass concentration according to the $\beta^*(z)$ signal few hours later and its development at neighboring stations did not change (Figure 10). As hygroscopic growth is included in the IFS-AER model but cloud formation by condensation nuclei is not, this process may significantly distort average biases of $\beta^*(z)$ during SDE as well as precipitation and radiation transfer (indirect aerosol effect) in the model. It further illustrates how errors may be

introduced by conversions of the primary model parameters (mass mixing ratio) to observed $\beta^*(z)$ . On 17 Oct 2017 also biomass burning aerosol released by forest fires in the north of Portugal was observed over north Germany as a shallow layer descending from initially 8-10 km ($\sim$3 UTC at Putbus) to 4-5 km altitude around noon. Observed $\beta^*(z)$ range from 0.1-1 $\cdot\ 10^{-5}Mm^{-1}sr^{-1}$. At the time of incipient cloud formation this layer still was clearly separated from the Saharan dust

layer below and thus could not influence this process.





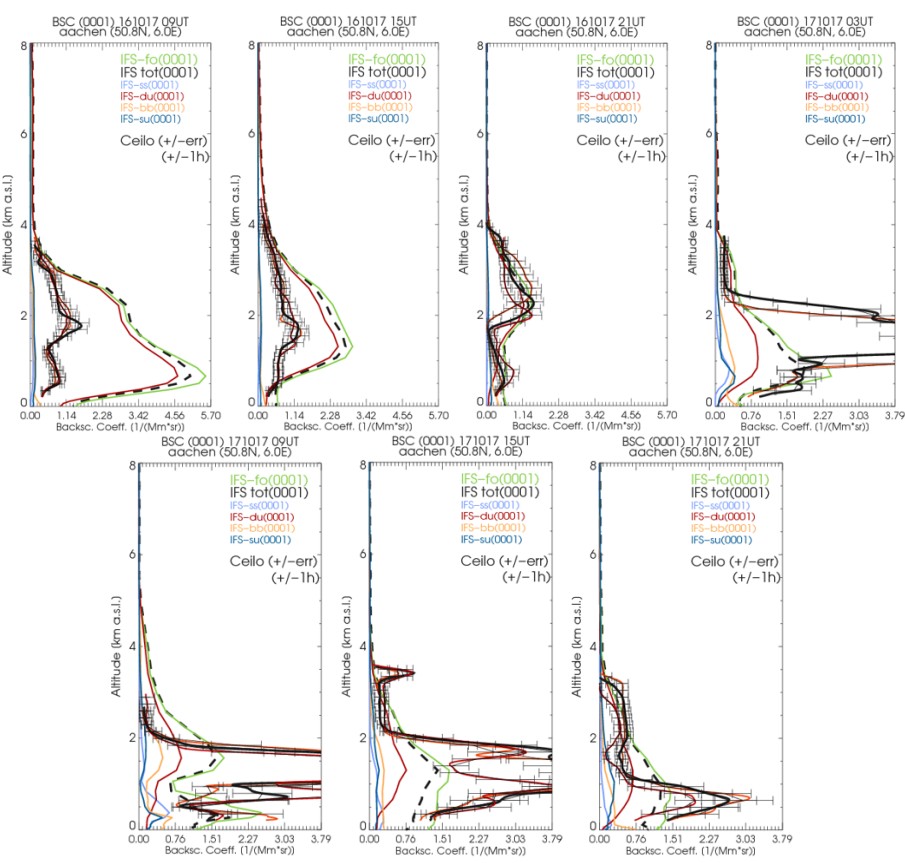

**Fig. 10.** Profiles of $\beta^*(z)$ from the ceilometer near Aachen on 16 (9, 15, 21 UTC) and 17 (3, 9, 15, 21 UTC) Oct 2017 from IFS-AER and ceilometer. The black dashed line is calculated with the DWD forward operator (FO), the green line using the ECMWF FO, retrieved as 'attenuated backscatter from ground' from the MARS archive. Onset of cloud formation occurs in the SD air mass on early 17 Oct. Colored profiles show the contributions of individual aerosol types.





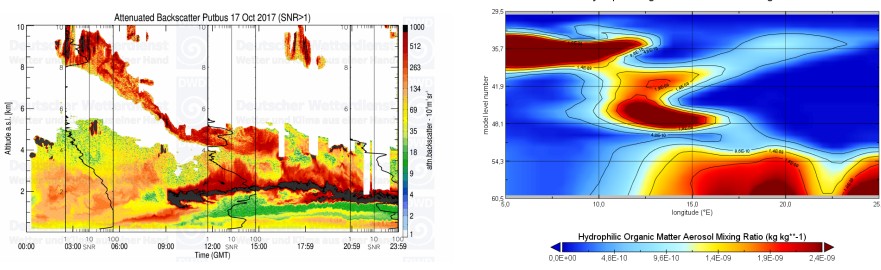

**Fig. 11.** Left: time-height section of $\beta^*(z)$ at the ceilometer station near Putbus on 17 Oct 2017. Right: Hydrophilic organic matter aerosol mass mixing ratio along 54.5°meridian simulated by IFS-AER on 17 Oct 2017, 15 UTC (plot generated with 'Panoply', provided by www.giss.nasa.gov/tools/panoply).

### 3.4.3 Fractions skill score

The penalizing of slightly vertically displaced aerosol layers yielding a low or even anti-correlation in Section 3.2 hints to the fact that a useful assessment of the positioning (in space and time) of an aerosol plume requires not only reference to point locations but also to their vicinity. Such a skill score shall distinguish nearly correct positioned features from deviations by a bigger margin. An ap-
proach to quantify the degree of overlap of simulated and observed aerosol structures is the fractions skill score (FSS; Roberts (2008); Skok and Roberts (2016)). The perceived accuracy increases with larger scales, longer averaging, elimination of outliers etc. Thus reasonable scales must be analysed to balance the processes of interest and the useful level of detail to be notified. For example small
(sub-grid) scale structures appear randomly displaced or missed because the information content of the model fields does not match the resolution of the observations, which the other way round, are not representative for the model grid box. For profile correlation, the usefulness-threshold of scales is for IFS-AER presently of the order of 1/2 day and 100 km. An approach towards FSS would be to adapt polygons to Figure 8, either outlining the boundary of an individual SD plume observed at
a given time at different ceilometer stations or, alternatively, refer to the overlap of plumes in time-height sections at individual stations. Another metric to quantify the model performance for coherent plumes in a quasi-stationary flow is the relative deviation of arrival/departure times of plumes/layers at station positions in model and observation as visualized in Figure 12 for the SD plume on 16 Oct 2017. Composite bullets with color coded arrival times observed in 2-D $\beta^*(z)$ ceilometer sections
(outer ring) and corresponding model fields (inner bullet) illustrate slightly delayed arrival (0-1 hour) of the model plume in western Germany, it's catch-up in the middle and again lagged arrival (0-2 hours) in the eastern part. The accuracy of determination is about one hour. This plume was neither observed nor simulated in the very south of Germany.





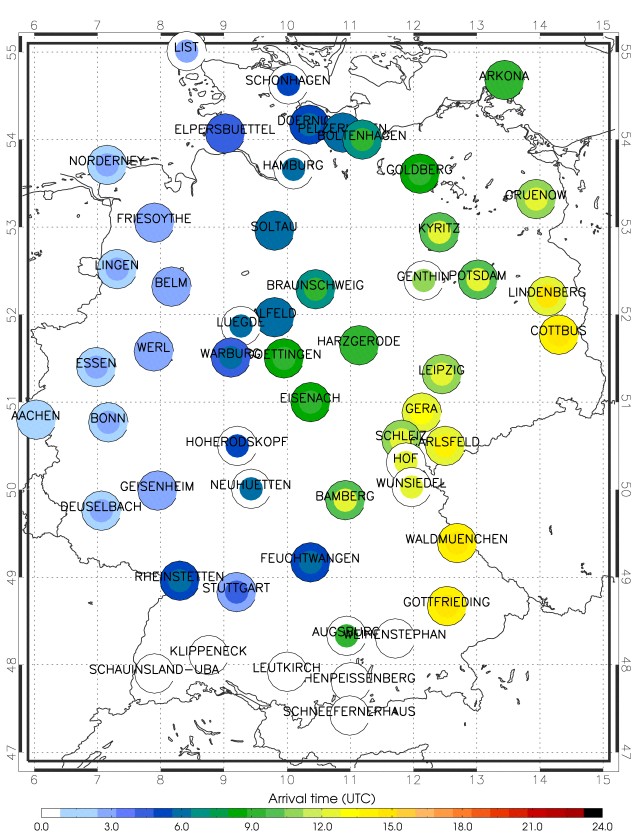

**Fig. 12.** Arrival time of individual SD plume on 16 Oct 2017, color-coded by the hour of day, as measured by ceilometer (outer ring) and IFS-AER (inner bullet). Missing data is white: the selected plume did not reach the southernmost part of Germany or arrival could not be identified due to low clouds.

## 4 Discussion and conclusions

Thorough evaluations of IFS-AER operational forecasts are regularly published in near-real-time or in retrospective validation reports on the CAMS website (http://macc-raq-op.meteo.fr/, https://atmosphere.copernicus.eu/eqa-reports-global-services/, last access: Sept 2020), as presented by Eskes et al. (2015). In these, the realism of the vertical profile has not yet received much attention,





although it may be relevant for aircraft guidance around volcanic ash layers, in cases of visibility
reduction during Saharan dust events, for the cloud formation potential or the dispersion of severe
pollution events. Our focus on the vertical aerosol distribution complements evaluations based on
AOD columns (Rémy et al., 2019; Gueymard and Yang, 2020) and surface in-situ measurements
(e.g. https://atmosphere.copernicus.eu/index.php/regional-services/). It extends our study by Chan
et al. (2018) CH18 from the surface layer up to the mid-troposphere. Yet, our results are shown to
be consistent with previous verifications. The vertical profile of $\beta^*(z)$ bias allows a more detailed
understanding of the height dependence of sources and sinks, vertical transport and re-distribution
of particles as well as temporal shifts in the model as FT biases are governed by long-range transport
rather than by surface drag and convection in the mixing layer. Verifications of IFS-AER re-analyses
reported by Flemming et al. (2017); Inness et al. (2019); Wagner et al. (2020) are in many respects
representative for the operational forecasts.

Compared to our first study by CH18, covering the period 09/2015-08/2016, we use the 4 years
period 2016-2019 with an overlap of 8 months, use data from 21 instead of 12 ceilometer stations
and all evaluable altitudes. As no clear dependence of performance on the distance to model grid
points was found in CH18, and the spatial resolution was increased from T255/L60 ($\approx 1°$x 1°) to
T511/L60 ($\approx 0.5°$x 0.5°) during cycle41r1 in July 2016, we drop the constraint to stations within
20 km around model grid points. Again $\beta^*(z)$ is used rather than backscatter coefficients as this is
the primary measured quantity of ceilometers that can be rigorously calculated from the IFS-AER
output. The small high-bias with large standard deviation of 1.5 times the model average found
by CH18 for near-surface-integrated (0.2-1 km altitude) $\beta^*(z)$ is confirmed by our analysis at the
lowest selected levels at 0.4 and 1 km a.g., as listed in Table 3. The larger overestimation of $\beta^*(z)$
associated with higher sea salt relative contributions is in CH18 partly ($\sim 10\%$ of total $\beta^*(z)$ ) at-
tributed to the utilized hygroscopic growth scheme OPAC (Hess et al., 1998), and is not elaborated
further in this study. Sea salt over continental Europe remains considerably overestimated (c.f Sec-
tion 3.3) in all seasons as changes to the sea salt emission scheme, e.g. coming in with cycle45r1
(06/2018) still primarily aim at decreasing the global low-bias of sea salt abundance dominated by
oceans. Concurrent substantial increases of sea-salt particle sizes and sinks (wet deposition) likely
reduce sea salt mass concentrations further inland, apparent as step at HPB in Figure 7, but are either
not efficient enough or still not the governing processes. As before, underestimated near-surface
$\beta^*(z)$ are presumably linked to unresolved local (Chan et al., 2018) or regional scale (e.g. Jan/Feb
2017) emission events and are often found to reach up to 2-3 km altitude (Figure 2). These emissions
are stronger and more frequent in winter and contribute to the bias/MNMB seasonal cycle (Figure 4).

Anti-correlated variances at low and high altitudes in ASM and CTR like in Apr16, Jun17,
Jan & Apr18 (Figures 3 and 2) attract attention, because they cancel out in vertically integrated





backscatter (Figure 5) and would likewise do so in AOD (not shown). They are often caused by vertical displacement of Saharan dust layers and are not reduced by the assimilation. As described by Benedetti et al. (2009) the IFS-AER 4D-Var assimilation scheme based on AOD columns could add vertical aerosol information by adaptions to the vertical temperature-, humidity- or wind profile, but the effect e.g by optimizing wind shears seems not specific enough to improve the simulated aerosol profile in ASM relative to CTR. Noticeable albeit not specific adaptions by the assimilation are evident during pollution or SD events as clear steps of the aerosol distribution with new analyses. At least some bias reduction by the assimilation is evident near the surface during severe anthropogenic pollution events in Jan/Feb 2017 and Feb/Mar 2018 (0.4 km line in Figure 2 and OM, BC, $SO_4$ in Figure 7). The scales of these events seem sufficiently large to produce significant updates during the assimilation step. Similarly low MNMB in $PM_{10}$ was found for the same event by Rémy et al. (2019) in their evaluation of cycle 45r1 without assimilation.

Performance changes of IFS-AER with height arise from different governing processes. As most sources enter into the lowest model level they have the largest effect to the lowest part of the profile. In the near-surface layer, the observed high-bias of $NO_3$ mass concentrations results from too efficient gas-to-particle partitioning, i.e. fine-mode $NO_3$ production from $HNO_3$-neutralisation by $NH_3$, followed by temperature dependent dissociation to $NO_3$ and $NH_4$ . Secondly, remaining $HNO_3$ may heterogeneously produce coarse $NO_3$ on SS or dust particles (Rémy et al., 2019), but this process is of minor relevance in central Europe where fine-mode nitrate has roughly 5 times larger mass concentration than coarse-mode nitrate. $NH_4$ is simulated at comparable concentrations as observed at HPB while $NO_3$ is about four times as high. For fine-mode $NO_3$ the most efficient sink near the surface, probably underestimated, is dry deposition (Zhang et al., 2012), while sedimentation of small particles should be slow and is disabled in the model. Below-cloud wet deposition (washout) should affect the whole profile rather than only the surface where the high-bias tendency toward the ground is found (Section 3.2). The increase of resolution from 1° to 0.5° in 06/2016 excluded Munich from the HPB grid box, which may contribute to the marked decrease of $PM_{10}$ and $SO_4$ around this time as in Figure 7. Since then it should be representative of HPB including only small surrounding towns and rural area. A particular value of the assessment w.r.t. mass lies in its independence from hygroscopic growth with humidity and any mass-to-optical conversions, which have particularly large impact on $SO_4$ and OM (Hong et al., 2014). The general bias increase towards the surface evident in Figure 3 may be caused by too slow vertical transport of surface emissions along with overestimated sources. $SO_4$ is overestimated in ASM in summer while typical central European surface concentrations in winter are met (Figure 7). Together with dust this causes most of the bias' seasonal cycle in Figure 4. The reason for the worsening mass input at surface level by the assimilation is not clear. OM is few $\mu g/m^3$ too high during all seasons since 02/2017. BC shows a step down with cycle 43r1 in 02/2017 and (except Jan/Feb 2017) a further downward tendency till 2019 at realistic





concentrations in ASM. Emission inventories thus seem to capture the decrease of anthropogenic emissions during the last decade but as for $SO_4$ the assimilation seems to add too much mass and may disturb the realistic partitioning between anthropogenic and biogenic OM.

Mixing layer: Against overestimation of mass concentrations and $\beta^*(z)$ near the surface, the aerosol load in the ML tends to be low-biased. This may stem from delayed vertical transport from the surface. Several more reasons may contribute to the dominance of low-biases of $\beta^*(z)$ in the ML. The forward operator, including mass-to-volume conversion, presently uses particle densities of the pure materials, not taking into account possible porosity of dry atmospheric particles enclosing air due to coagulation and variable internal mixing (Winkler et al., 1981). If the model assumed larger bulk densities than particles actually have, the equivalent volumes were calculated too small and optical properties would be underestimated, because they depend strongly on the particle size. The density of accumulation mode particles, composed of hydrophilic and hydrophobic materials could be overestimated by up to a factor 1.5, which transfers to a factor 1.3 in the optically relevant surface area. Secondly, the ML top is too smooth which means the capping transport barrier at the ML top seems less effective in the model, diluting higher ML concentrations with cleaner FT air. Geometrically, however, the ML height on average seems reasonable (cf. Section 4.1).

The monthly mean bsc profiles suggest that aerosol mass, added to the whole column by the assimilation (runs 0001), results in overall higher aerosol load than in the control runs till 07/2019, but the assimilation does not improve the step at the top of the PBL to lower values in the free troposphere (FT) as shown by the ceilometer profiles. Note that daily/monthly averaging considerably smooths the ML top due to daily/monthly variations of PBL height. Likewise, the FT background might be slightly high-biased due to adding an assimilated portion there. This aerosol mass would be missing in the ML, yielding a too low amplitude (coded in the standard deviation) of the model compared to observations (reference) in the Taylor plots, too.

Free troposphere: Saharan dust, as an irregular constituent is (as in CH18) found to be over-estimated over Germany by typically a factor 2 or more all the time (except when water condenses to the dust). CH18 calculated, that accounting for non-spherical particles, using conversion coefficients based on T-Matrix calculations rather than Mie-Theory, would reduce $\beta^*(z)$ by 15-45%. This reduction arises from the modification of the phase function by non-sphericity, coded in the lidar ratio LR, not the specific extinction, and thus does not transfer to AOD. In order to reduce the dust high bias in the model, the dust source size distribution after cycle 43r1 was modified to distribute less mass into the fine (8 to 5%) and more mass into the super-coarse bin (61 to 83%) which has a shorter lifetime due to faster sedimentation (Rémy et al., 2019). An according dust reduction can however not be seen over Germany in the mass concentrations of the dust-proxy in Figure 7, which is independent from uncertainties in the the mass-to-$\beta^*(z)$ conversion. That the observed inflation of the $\beta^*(z)$ signal on 17 Oct 2017 (Section 3.4.2) marks the onset of cloud formation, seems plausible.





On the one hand the temporary factor 10 increase of $\beta^*(z)$ ($\sim 1 \text{ x } 10^{-6} \rightarrow \sim 1.2 \text{ x } 10^{-5}$) roughly corresponds to a significant visual range reduction by an order of magnitude (to $\sim 1$ km assuming a lidar ratio near 30 sr). On the other hand a mature water cloud would block the lidar beam and any signal from above, which may have been observed at some stations further north (not shown). Alternatively the passage of a shallow plume with ten times higher concentration would have been

diagnosed which would be rather untypical. Unfortunately satellite imagery provides only blurred pictures due to optically dense smoke layers above. In all, the unambiguous identification of particle induced cloud formation is a challenge in the observations as well as it is for a model to simulate hygroscopic growth near saturation.

  The extended smoke layer that arrived on early 17 Oct 2017 few km above the dust plume in a strong

south-westerly flow around 8-10 km altitude from Portugal is clearly evident in TERRA/MODIS reflectance imagery (https://worldview.earthdata.nasa.gov/), in the northern German ceilometers and in organic matter fields of IFS-AER. With small vertical wind shear the simulated smoke curtain tilted downward by $\sim 2°$ lat/lon from NW (8-10 km) to SE (<2 km) and was passively advected north-eastward across north Germany. Its passage over the ceilometer stations is reproduced in de-

tail by IFS-AER (not shown), only that the observations show a 1 km thin streamer reaching down to 3.5 km, where it is too thick and reaches too far down in the model (2 km), likely due to the resolution (cf. Figure 11). The dynamic reason for the tilted smoke curtain may be ignored for our purpose, but the comparison confirms the expected behavior found for previous fire cases that IFS-AER forecasts are capable to reproduce many details of smoke plumes qualitatively, but that

the simulated shape and position become more uncertain when smaller scales develop (Kaiser et al., 2012). It shows that source strengths, injection heights and long-range transport in the model are quite realistic. But it must be noted, that $\beta^*(z)$ of the smoke plume is considerably underestimated. This may be due to the model resolution, but also to emission strengths and heights that are inferred from fire radiative power measurements onboard satellites and converted into convective updraft.

The smoke-mass is calculated from the thermal energy via land-cover-dependent conversion factors, which can represent local characteristics only to a certain degree (Kaiser et al., 2012).

## 4.1   Mixing layer height

  The mixing layer height MLH characterizes the ML in many respects, as it can be closely related to

important variables like water vapor, cloud cover, heat fluxes and vertical transport as well as $PM_{2.5}$ and contaminant dispersion (Engeln and Teixeira, 2013; Li et al., 2020). It is however challenging to infer operationally as described in Section 2.3 and by Haeffelin et al. (2012). The physical correspondence between observed aerosol gradients and turbulence (Ri > 0.25) requires meteorological conditions without vertical shear (generating vertical aerosol gradients), fog, clouds, precipitation or

aerosol plumes (often dust), which after rigorous filtering leaves only about 5-10% of the days. SD





alone already affects much more than 220 of 1461 days. Regarding the large uncertainties of MLH determination, the maximum daily MLH at ceilometer stations MMLH is used as a robust proxy with still significant information content. The correlation of (filtered) daily reported (CHM15k) and manually derived MMLH for the mid-German region around Alfeld (52.0° N, 9.8° E) is medium
to low (r = 0.31), including few outliers. The long-term agreement of model-diagnosed MMLH height from ECMWF's NWP model and visually derived MMLH is however remarkably good with strong correlation of r = 0.66 as shown in Figure 13. Medium-term MMLH fluctuations (time scale 1 month), associated to changes in the general circulation, are mostly captured. The MMLH is generally underestimated by the model, but the bias seems to decrease toward the end of the period.
In presence of tremendous variability, the underestimation by -200 m to -600 m in 2016 and 2017 seems to jump to 0 to -300 m with the change to cycle 43r3 in Okt 2017 and again concentrates at values between -200 m and -500 m during cycle 46r1 after July 2019. Again we emphasize that this discussion of MMLH is intended more as a case study rather than a rigorous evaluation, which is beyond the scope of this article. In Figure 14 a composite contour-bullet plot of IFS MLH with su-
perimposed station values from 20 Mar 2019 illustrates the behavior of MMLH for a calm day with undisturbed ML development over large parts of Germany. Here IFS-AER is capable to reproduce the NW-to-SE increase of the MMLH related to the transit from low to higher pressure. Few stations sticking out with specifically high or low MMLH probably exhibit local influences like heat-island effects over large cities (e.g. 'lei' = Leipzig) where the residual layer remained high and convective
over the night. Or over isolated mountains (e.g. Brocken/Torfhaus - the deep blue dot near 51° N, 10.5° E) the ML top may not follow the steep terrain and local MLG above ground will be too low. Thus this short study shows that rigid checks of station characteristics, data quality and outliers are necessary before operational MLH data from ceilometers (or lidars) may be used to constrain and evaluate models.






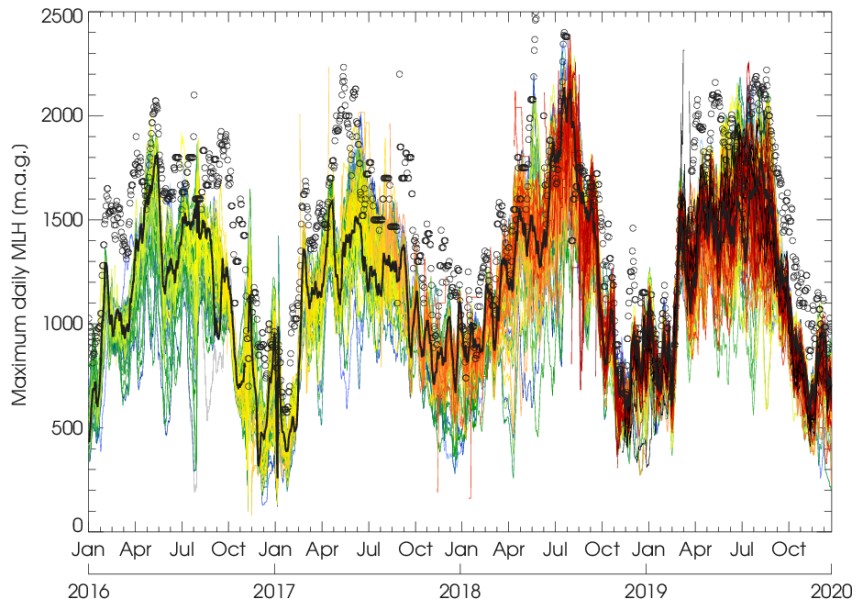

**Fig. 13.** Maximum daily mixing layer height a.g. (MMLH) observed at the position of the German ceilometer network stations and extracted from the ECMWF NWP model for the period 01/2016 - 12/2019. Different colors refer to model MMLH at different stations (Colors shift from green to red because the number of available stations increases over the years). The dashed and the solid black lines pick out the MMLH inferred for the region around the northern German station Alfeld (9.9 °E, 52.0 °N), manually from the daily ceilometer time-height plots and from the model fields, respectively.



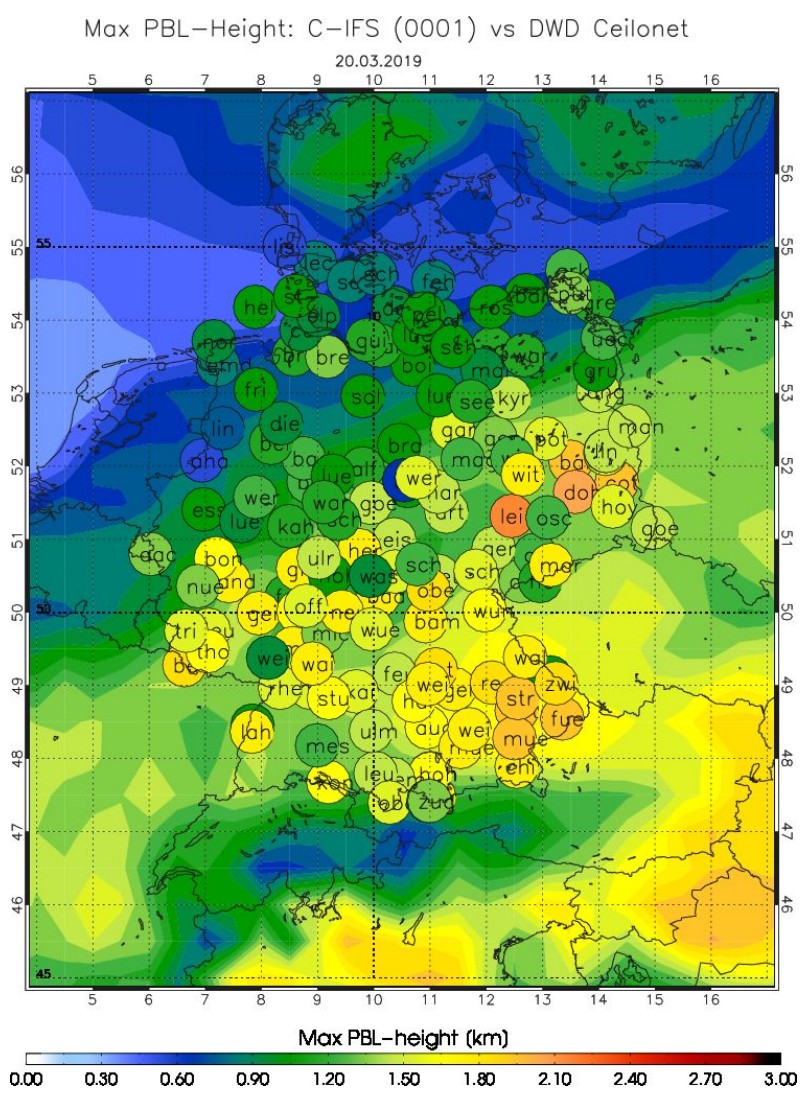

**Fig. 14.** Map of maximum daily MLH as simulated by IFS-AER (filled contours) with MMLH at German ceilometer network stations for 20 Mar 2019, overlayed as correspondingly color-coded bullets.

### 4.2 Uncertainties and limitations

The overall uncertainty of our results is mainly limited by the conversion of model mass mixing ratios to $\beta^*(z)$ , the uncertainty of the ceilometer observations, and the re-sampling over different horizontal and vertical resolutions. The former includes estimates of particle shape, densities, mixing- and hygroscopic state as well as meteorological conversions as described in Chan et al.





(2018) and updated in Section 2.1. The uncertainties inherent in the observations are discussed in Section 2.2. Of these, clearly the lower and upper altitude ranges of the profiles are affected by the incomplete overlap of laser-beam and receiver field-of-view and low signal-to-noise ratio SNR as well as contamination by clouds, respectively. At the lowest considered altitude 400 m a.g. the signal

is typically at $\sim$10-20% of it's full-overlap value, which can reasonably be corrected by a $\sin^2$-like step function. The crucial degradation of the SNR is by clouds, precipitation and the $r^{-2}$-decrease with distance. Within the evaluated range from 0.4 - 6 km, however, the combined uncertainty due to these contributions mostly is small compared to the model-observation biases in question. Owing to the typical half-daily time scale of transport precision (c.f. Section 3.2) also the distance from

the model grid points seems negligible. Many of the results, however, are sensitive to the applied scales, and some examples have been discussed where the increased horizontal (06/2018) and vertical (07/2019) resolution lead to better matches between observed and forecast structures. It has to be kept in mind that the relatively coarser global fields of the CAMS system are intended to serve as boundary conditions for nested regional models which refine the aerosol distributions down to scales

of few km.

As there was no $\beta^*(z)$ output available from IFS-AER before cycle45r1 (10/2017), we use for consistency over the whole period the same lidar forward operator to calculate $\beta^*(z)$ from the model mass mixing ratios as described in Chan et al. (2018). Minor modifications were necessary to integrate the higher resolution and additional species ($NO_3$, $NH_4$) as of 07/2019. It uses their pre-

calculated look-up table, slightly adapted to IFS-AER values and modified to additionally handle $NO_3$ and $NH_4$(cf. Table A1, A2, A3). Since 10/2017 lidar output is available from the IFS archive. Results from both emulators compare well for dust, but for other components like sea salt somewhat different $\beta^*(z)$ profiles are calculated. Possible reasons may be the handling of hygroscopic growth near saturation, the disregard of (however small) absorption by trace gases at 1064 nm and a differ-

ent effective resolution of the model fields resulting from both lidar emulators. A direct comparison of the $\beta^*(z)$ (from ground) product retrieved from the IFS and that calculated from the model mass mixing ratios according to (Chan et al., 2018) reveals that the IFS $\beta^*(z)$ product is provided with a different effective resolution then the mmr fields used here. The gradients thus appearing at the boundaries of aerosol structures cause deviating results depending on the specific time and location

of the comparison. For longer averages as mostly discussed in this article, these differences largely cancel out.

## 5 Summary

The assessment of IFS-AER vertical aerosol distributions with calibrated ceilometer profiles over

Germany (central Europe) generally confirms the realistic reproduction of the vertical aerosol vari-





ability in terms of attenuated backscatter $\beta^*(z)$. The shape of the profile, dominated by the mixing layer ML and occasionally by long-range transport particles is largely captured, as indicated by high co-variance of daily average profiles with Pearson's r $\approx$ $0.6 - 0.95$, however no clear impact of the assimilation is found. In summer the agreement of profile shapes is less due to small vertical shifts or untimely long-range transport to which r is quite sensible. A systematic higher/lower bias regularity is found in the lower part of the profile, i.e. by trend higher bias near the ground to lower bias (underestimation) in the middle ML aerosol load. It is attributed to over-estimated sources at the surface, likely in combination with too slow vertical transport and a probably too weak transport barrier at the top of the ML, where the large aerosol gradient is not fully captured. The low aerosol background in the free troposphere FT is usually reproduced. Also captured are plumes and layers from long-range transport of Saharan dust and fire smoke, although $\beta^*(z)$ of dust is overestimated over Germany by a factor 2 or more, and small scale structures evolving during the dispersion of these layers can not be resolved at the present model resolution.

Comparison to dry-state aerosol in-situ observations suggest that $SO_4$ and OM sources as well as gas-to-particle partitioning of the $NO_3$-$NH_4$-system are too strong, while black carbon load and trend is realistic near the surface. With respect to the discussed metrics, no consistent development is evident due to the five model upgrades during the evaluated period. The vertically integrated $\beta^*(z)$, which codes similar information like AOD, consistently with these previous findings shows a bias near zero for 43r1 (till 05/2016) and 46r1 (after 07/19) and slightly negative in-between. The modified normalized mean bias MNMB which is less dependent on absolute values reveals lower values in the more relevant (for air-quality) surface- and mixing layer and a general increase toward higher levels. Over the whole period, the bias of $\beta^*(z)$ exhibits seasonal cycles at the lower levels due to overestimation of $SO_4$ and OM sources/lifetimes in summer and under-representation of severe pollution episodes in winter.

Finally, we demonstrated that ceilometer networks offer several options to check the realism of mixing layer heights in atmospheric numerical models. Though we confined to manual analysis of a representative region, we could provide confidence that the annual cycle and the maximum daily height of the ML can be reproduced within few 100 m vertically by the IFS-AER model.

In future, the regional extension of this assessment to larger parts of Europe and the combination of ceilometer networks' spatio-temporal coverage with the higher accuracy and particle identification capability of sun-photometers (AERONET) and multi-wavelength-depolarisation (Raman-) lidars will significantly reduce the uncertainties remaining in this study. Corresponding CAMS activities, also for evaluation of the particle composition using European network data have already started. A robust discussion of boundary layer heights will benefit more from further improvements to the algorithms than from improved profile data quality.





*Data availability* The source code of the ECMWF IFS model is not available for public as it is intellectual property of the ECMWF and its member states. ECMWF IFS model simulation results are available to the meteorological offices of the ECMWF member states. The ceilometer raw data are available on request from the data originator DWD (datenservice@dwd.de). GAW in-situ data are available from the EBAS data centre at http://ebas.nilu.no/. The database of aerosol optical properties used in this study is available on request from the corresponding author (harald.flentje@dwd.de).

*Author contribution* IM analysed the ceilometer data. SR and ZK are involved in IFS-AER development, provided model information and interpretations and aided in the retrieval of IFS-AER data. WT organized the DWD ceilometer network and contributed to data transmission. HF performed the evaluation and prepared the manuscript with contributions from all co-authors.

*Competing interests* The authors declare that they have no conflict of interest.

*Acknowledgements.* The Copernicus Atmosphere Monitoring Service (CAMS) is hosted by the European Centre for Medium-Range Weather Forecasts (ECMWF) since 10/2015. The 'Global and regional a posteriori evaluation and quality assurance' (CAMS 84) is funded by the European Union under the Framework Agreement 2015/CAMS-84 through its main contractor Royal Netherlands Meteorological Institute (KNMI). The work for this article was supported by the European Cooperation in Science and Technology (COST) action 'ToProf' of the European Union's Horizon 2020 programme (Project No. ES1303), followed by E-PROFILE (e-profile.eu) which is part of the EUMETNET Composite Observing System EUCOS. We thank Maxime Hervo and his colleagues at MeteoSwiss for the provision of the ToProf E-Profile Rayleigh calibration routine, NASA and NOAA for making MODIS imagery and the HYSPLIT model publicly available. The Deutscher Wetterdienst DWD operates the German ceilometer network and the Hohenpeißenberg observatory as a global station in the WMO Global Atmosphere Watch program (https://community.wmo.int/activity-areas/gaw).





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





## Appendix A  Bias time series

For clarification to Figure 2 the 1-d and 30-d average figures are shown to illustrate the variability
and the longer-term variation of the somewhat unclear overlaying lines for the different altitudes.

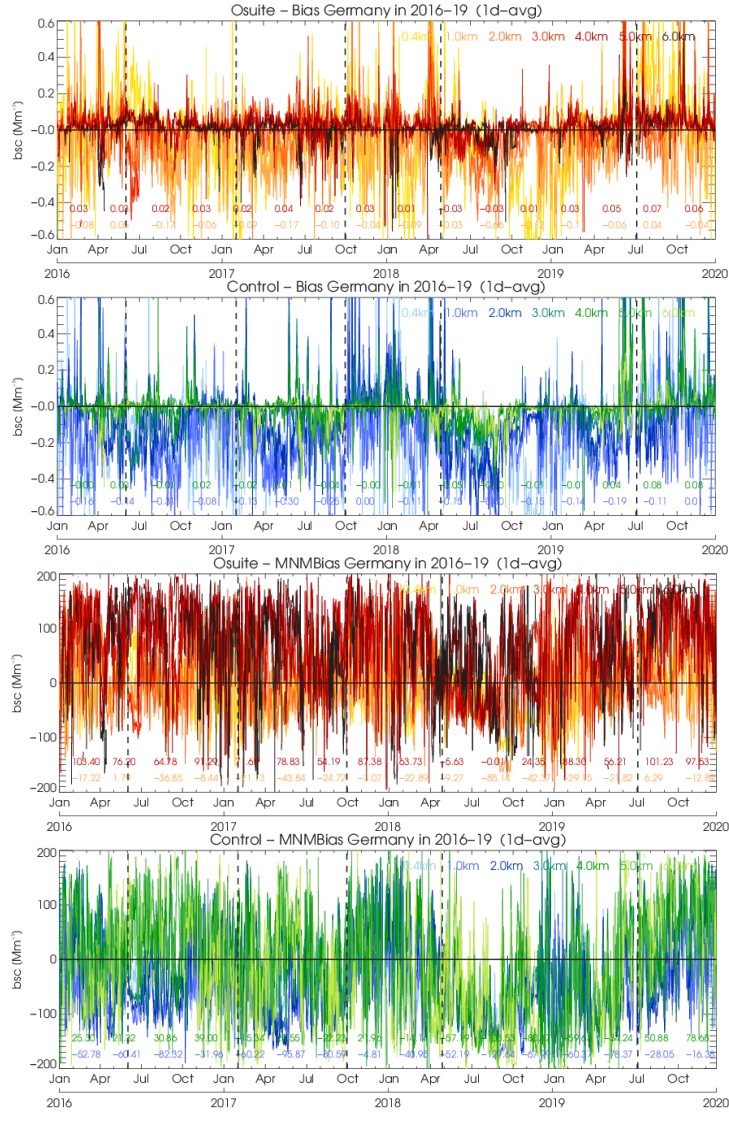

**Fig. A1a.** 1-day running mean bias of $\beta^*(z)$ from ASM (1st panel) and CTR (2nd panel) combined from 21
German stations in 2016-2019. Same for modified normalized mean bias (MNMB) in 3rd and 4th panel. Colors
refer to different altitudes above ground. Vertical black lines indicate major model updates as in Table 1.

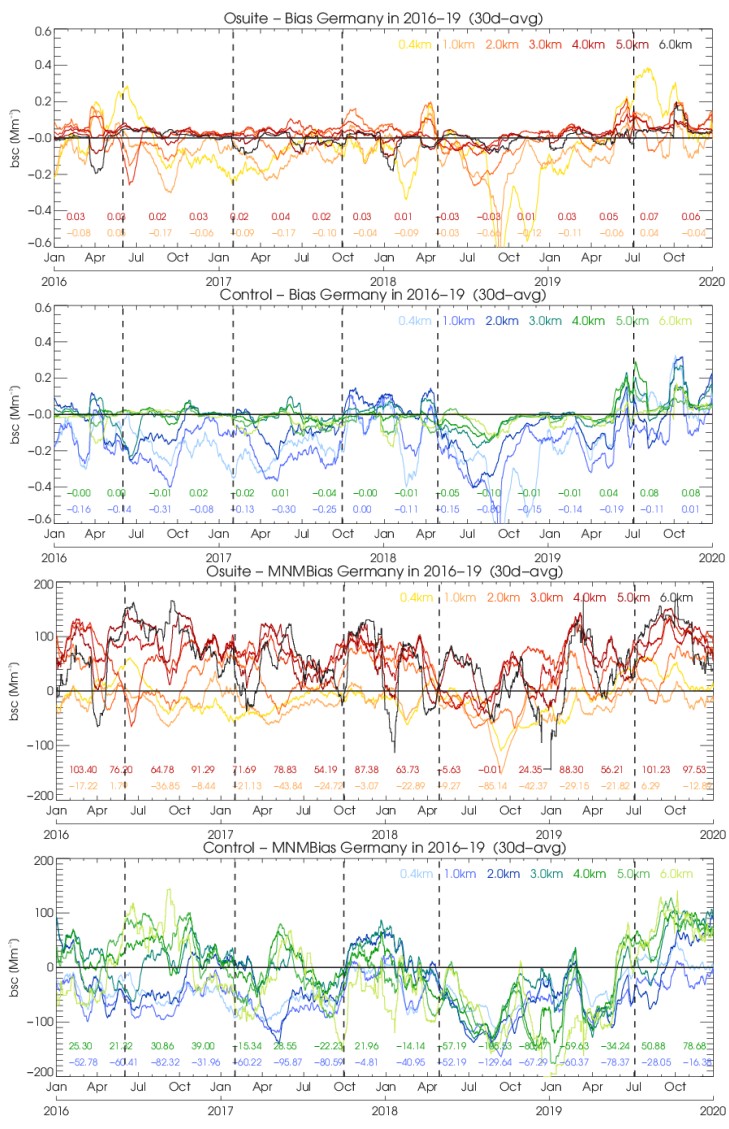

**Fig. A1b.** 30-day running mean bias of $\beta^*(z)$ from ASM (1st panel) and CTR (2nd panel) combined from 21 German stations in 2016-2019. Same for modified normalized mean bias (MNMB) in 3rd and 4th panel. Colors refer to different altitudes above ground. Vertical black lines indicate major model updates as in Table 1. The dips in the 6 km bias around Jan 2018 and Jan 2019 are caused by edge truncation when averaging over sparse (cloudy) data.





### Appendix B  Look-up-tables for forward operator

The most important aerosol optical and micro-physical properties used for converting model mass mixing ratios mmr to attenuated backscatter $\beta^*(z)$ (the 'forward operator' or 'lidar emulator') are 1270 listed in the following tables. All values are valid for the near-infrared 1064 nm wavelength used by the CHM15k/CHM15kx ceilometers of the German Meteorological Service's operational network.



**Table A1.** Microphysical properties of aerosols assumed for conversion of model mass mixing ratios to $\beta^*(z)$ at 1064 nm.

| Aerosol Type | Rel Humidity % | Density ($\varrho_p$, g/cm$^3$) | Grow Factor | Spec. Ext. Cross Section $\sigma_e^*$ (m$^2$/g) | Lidar Ratio $S_p$ (sr) | Single Scattering Albedo ($\omega_0$) |
|---|---|---|---|---|---|---|
| Sea Salt(0.03-0.5) | 0 | 2.160 | 1.00 | 0.127 | 21.72 | 0.998902 |
| Sea Salt(0.03-0.5) | 10 | 1.821 | 1.12 | 0.127 | 21.72 | 0.998959 |
| Sea Salt(0.03-0.5) | 20 | 1.603 | 1.24 | 0.127 | 21.72 | 0.998983 |
| Sea Salt(0.03-0.5) | 30 | 1.455 | 1.37 | 0.127 | 21.72 | 0.99899 |
| Sea Salt(0.03-0.5) | 40 | 1.352 | 1.49 | 0.810 | 56.33 | 0.998968 |
| Sea Salt(0.03-0.5) | 50 | 1.278 | 1.61 | 1.146 | 56.94 | 0.999596 |
| Sea Salt(0.03-0.5) | 60 | 1.232 | 1.71 | 1.542 | 58.65 | 0.999659 |
| Sea Salt(0.03-0.5) | 70 | 1.196 | 1.81 | 2.140 | 65.49 | 0.999717 |
| Sea Salt(0.03-0.5) | 80 | 1.147 | 1.99 | 3.234 | 75.81 | 0.999779 |
| Sea Salt(0.03-0.5) | 85 | 1.111 | 2.19 | 3.878 | 76.0978 | 0.999807 |
| Sea Salt(0.03-0.5) | 90 | 1.086 | 2.38 | 4.862 | 73.3724 | 0.999846 |
| Sea Salt(0.03-0.5) | 95 | 1.047 | 2.91 | 9.632 | 78.4961 | 0.99989 |
| Sea Salt(0.5-5) | 0 | 2.160 | 1.00 | 0.145 | 10.1023 | 0.992657 |
| Sea Salt(0.5-5) | 10 | 1.821 | 1.12 | 0.145 | 10.1023 | 0.991804 |
| Sea Salt(0.5-5) | 20 | 1.603 | 1.24 | 0.145 | 10.1023 | 0.990984 |
| Sea Salt(0.5-5) | 30 | 1.455 | 1.37 | 0.145 | 10.1023 | 0.990086 |
| Sea Salt(0.5-5) | 40 | 1.352 | 1.49 | 0.302 | 13.7809 | 0.989224 |
| Sea Salt(0.5-5) | 50 | 1.278 | 1.61 | 0.354 | 14.3385 | 0.995823 |
| Sea Salt(0.5-5) | 60 | 1.232 | 1.71 | 0.407 | 14.748 | 0.996317 |
| Sea Salt(0.5-5) | 70 | 1.196 | 1.81 | 0.470 | 14.7443 | 0.996842 |
| Sea Salt(0.5-5) | 80 | 1.147 | 1.99 | 0.570 | 14.6123 | 0.997375 |
| Sea Salt(0.5-5) | 85 | 1.111 | 2.19 | 0.651 | 15.1343 | 0.997644 |
| Sea Salt(0.5-5) | 90 | 1.086 | 2.38 | 0.792 | 18.6968 | 0.998097 |
| Sea Salt(0.5-5) | 95 | 1.047 | 2.91 | 1.140 | 15.678 | 0.998713 |
| Sea Salt(5-20) | 0 | 2.160 | 1.00 | 0.041 | 18.2163 | 0.978392 |
| Sea Salt(5-20) | 10 | 1.821 | 1.12 | 0.041 | 18.2163 | 0.976231 |
| Sea Salt(5-20) | 20 | 1.603 | 1.24 | 0.041 | 18.2163 | 0.973844 |
| Sea Salt(5-20) | 30 | 1.455 | 1.37 | 0.041 | 18.2163 | 0.971703 |
| Sea Salt(5-20) | 40 | 1.352 | 1.49 | 0.082 | 14.3399 | 0.969431 |
| Sea Salt(5-20) | 50 | 1.278 | 1.61 | 0.095 | 14.3044 | 0.987793 |
| Sea Salt(5-20) | 60 | 1.232 | 1.71 | 0.108 | 14.4325 | 0.989233 |
| Sea Salt(5-20) | 70 | 1.196 | 1.81 | 0.127 | 14.8442 | 0.990821 |
| Sea Salt(5-20) | 80 | 1.147 | 1.99 | 0.153 | 15.3336 | 0.992415 |
| Sea Salt(5-20) | 85 | 1.111 | 2.19 | 0.175 | 17.2092 | 0.993225 |
| Sea Salt(5-20) | 90 | 1.086 | 2.38 | 0.214 | 9.5161 | 0.994551 |
| Sea Salt(5-20) | 95 | 1.047 | 2.91 | 0.316 | 8.2696 | 0.996283 |
| Dust(0.03-0.55) | 0 | 2.610 | 1.00 | 1.496 | 78.5535 | 0.996971 |
| Dust(0.55-0.9) | 0 | 2.610 | 1.00 | 1.611 | 48.6388 | 0.996741 |
| Dust(0.9-20) | 0 | 2.610 | 1.00 | 0.445 | 13.3959 | 0.987986 |

[a] Sea salt aerosols are represented in the model by three size bins with bin limits set to 0.03-0.5 $\mu$m (bin 1), 0.5-5 $\mu$m (bin 2) and 5-20 $\mu$m (bin 3). [b] Dust aerosols are represented in the model by three size bins with bin limits set to 0.03-0.55 $\mu$m (bin 1), 0.55-0.90 $\mu$m (bin 2) and 0.90-20.00 $\mu$m (bin 3). [c] A bimodal log-normal size distribution is assumed for sea salt aerosols, with $r_0$=0.1002 $\mu$m and 1.002 $\mu$m and $\sigma_g$=1.9 and 2.0. A monomodal size distribution is assumed for dust. The number concentrations $N_1$ and $N_2$ of the first and second mode are 70 and 3 cm$^{-1}$, respectively. Note that density of hydrophilic aerosol changes with hygroscopic growth of particle.





**Table A2.** Microphysical properties of aerosols assumed for conversion of model mass mixing ratios to $\beta^*(z)$ at 1064 nm.

| Aerosol Type | Rel Humidity % | Density $(\varrho_p, \text{g/cm}^3)$ | Grow Factor | Spec. Ext. Cross Section $\sigma_e^*$ (m²/g) | Lidar Ratio $S_P$ (sr) | Single Scattering Albedo ($\omega_0$) |
|---|---|---|---|---|---|---|
| Organic Matter (hydrophobic) | 0 | 1.769 | 1.00 | 0.768 | 34.15 | 1 |
| Organic Matter (hydrophobic) | 10 | 1.769 | 1.00 | 0.768 | 34.15 | 1 |
| Organic Matter (hydrophobic) | 20 | 1.769 | 1.00 | 0.768 | 34.15 | 1 |
| Organic Matter (hydrophobic) | 30 | 1.769 | 1.00 | 0.768 | 34.15 | 1 |
| Organic Matter (hydrophobic) | 40 | 1.769 | 1.00 | 0.768 | 34.15 | 1 |
| Organic Matter (hydrophobic) | 50 | 1.769 | 1.00 | 0.768 | 34.15 | 1 |
| Organic Matter (hydrophobic) | 60 | 1.769 | 1.00 | 0.768 | 34.15 | 1 |
| Organic Matter (hydrophobic) | 70 | 1.769 | 1.00 | 0.768 | 34.15 | 1 |
| Organic Matter (hydrophobic) | 80 | 1.769 | 1.00 | 0.768 | 34.15 | 1 |
| Organic Matter (hydrophobic) | 85 | 1.769 | 1.00 | 0.768 | 34.15 | 1 |
| Organic Matter (hydrophobic) | 90 | 1.769 | 1.00 | 0.768 | 34.15 | 1 |
| Organic Matter (hydrophobic) | 95 | 1.769 | 1.00 | 0.768 | 34.15 | 1 |
| Organic Matter (hydrophilic) | 0 | 1.769 | 1.00 | 0.768 | 34.15 | 1 |
| Organic Matter (hydrophilic) | 10 | 1.607 | 1.08 | 0.768 | 34.15 | 1 |
| Organic Matter (hydrophilic) | 20 | 1.488 | 1.16 | 0.768 | 34.15 | 1 |
| Organic Matter (hydrophilic) | 30 | 1.397 | 1.25 | 0.768 | 34.15 | 1 |
| Organic Matter (hydrophilic) | 40 | 1.328 | 1.33 | 1.112 | 39.78 | 1 |
| Organic Matter (hydrophilic) | 50 | 1.274 | 1.41 | 1.289 | 41.33 | 1 |
| Organic Matter (hydrophilic) | 60 | 1.233 | 1.49 | 1.531 | 43.22 | 1 |
| Organic Matter (hydrophilic) | 70 | 1.199 | 1.57 | 1.891 | 45.71 | 1 |
| Organic Matter (hydrophilic) | 80 | 1.157 | 1.70 | 2.542 | 49.47 | 1 |
| Organic Matter (hydrophilic) | 85 | 1.128 | 1.82 | 3.158 | 52.398 | 1 |
| Organic Matter (hydrophilic) | 90 | 1.105 | 1.94 | 4.329 | 56.95 | 1 |
| Organic Matter (hydrophilic) | 95 | 1.065 | 2.27 | 8.267 | 66.875 | 1 |
| Black Carbon (hydrophobic) | 0 | 1.000 | 1.00 | 3.898 | 168.265 | 0.0837982 |
| Black Carbon (hydrophilic) | 0 | 1.000 | 1.00 | 3.898 | 168.265 | 0.0837982 |
| Sulfate | 0 | 1.769 | 1.00 | 1.060 | 34.14 | 1 |
| Sulfate | 10 | 1.769 | 1.08 | 1.060 | 34.14 | 1 |
| Sulfate | 20 | 1.769 | 1.16 | 1.060 | 34.14 | 1 |
| Sulfate | 30 | 1.769 | 1.25 | 1.060 | 34.14 | 1 |
| Sulfate | 40 | 1.430 | 1.33 | 1.540 | 39.75 | 1 |
| Sulfate | 50 | 1.390 | 1.41 | 1.783 | 41.29 | 1 |
| Sulfate | 60 | 1.349 | 1.49 | 2.117 | 43.18 | 1 |
| Sulfate | 70 | 1.302 | 1.57 | 2.615 | 45.658 | 1 |
| Sulfate | 80 | 1.245 | 1.70 | 3.516 | 49.394 | 1 |
| Sulfate | 85 | 1.210 | 1.82 | 4.368 | 52.311 | 1 |
| Sulfate | 90 | 1.165 | 1.94 | 5.988 | 56.839 | 1 |
| Sulfate | 95 | 1.101 | 2.27 | 11.436 | 66.8957 | 1 |



**Table A3.** Microphysical properties of aerosols assumed for conversion of model mass mixing ratios to $\beta^*(z)$ at 1064 nm.

| Aerosol Type | Rel Humidity % | Density ($\varrho_p$, g/cm$^3$) | Grow Factor | Spec. Ext. Cross Section $\sigma_e^*$ (m$^2$/g) | Lidar Ratio $S_p$ (sr) | Single Scattering Albedo ($\omega_0$) |
|---|---|---|---|---|---|---|
| Nitrate(fine) | 0 | 1.769 | 1.00 | 0.232 | 33.5 | 1 |
| Nitrate(fine) | 10 | 1.769 | 1.00 | 0.232 | 33.5 | 1 |
| Nitrate(fine) | 20 | 1.769 | 1.00 | 0.232 | 33.5 | 1 |
| Nitrate(fine) | 30 | 1.769 | 1.00 | 0.232 | 33.5 | 1 |
| Nitrate(fine) | 40 | 1.430 | 1.10 | 0.351 | 36.3 | 1 |
| Nitrate(fine) | 50 | 1.390 | 1.20 | 0.412 | 39.3 | 1 |
| Nitrate(fine) | 60 | 1.349 | 1.25 | 0.498 | 40.8 | 1 |
| Nitrate(fine) | 70 | 1.302 | 1.30 | 0.632 | 42.3 | 1 |
| Nitrate(fine) | 80 | 1.245 | 1.35 | 0.895 | 43.9 | 1 |
| Nitrate(fine) | 85 | 1.210 | 1.50 | 1.097 | 48.5 | 1 |
| Nitrate(fine) | 90 | 1.165 | 1.70 | 1.518 | 54.8 | 1 |
| Nitrate(fine) | 95 | 1.101 | 2.10 | 3.121 | 66.4 | 1 |
| Nitrate(coarse) | 0 | 1.769 | 1.00 | 0.355 | 18 | 1 |
| Nitrate(coarse) | 10 | 1.769 | 1.00 | 0.355 | 12.6 | 1 |
| Nitrate(coarse) | 20 | 1.769 | 1.00 | 0.355 | 11.3 | 1 |
| Nitrate(coarse) | 30 | 1.769 | 1.00 | 0.355 | 11.3 | 1 |
| Nitrate(coarse) | 40 | 1.430 | 1.10 | 0.443 | 11.9 | 1 |
| Nitrate(coarse) | 50 | 1.390 | 1.20 | 0.555 | 12.6 | 1 |
| Nitrate(coarse) | 60 | 1.349 | 1.25 | 0.623 | 14.1 | 1 |
| Nitrate(coarse) | 70 | 1.302 | 1.30 | 0.697 | 15.9 | 1 |
| Nitrate(coarse) | 80 | 1.245 | 1.35 | 0.780 | 17.1 | 1 |
| Nitrate(coarse) | 85 | 1.210 | 1.50 | 1.093 | 18 | 1 |
| Nitrate(coarse) | 90 | 1.165 | 1.70 | 1.682 | 19 | 1 |
| Nitrate(coarse) | 95 | 1.101 | 2.10 | 3.651 | 18.7 | 1 |
| Ammonium | 0 | 1.769 | 1.00 | 0.212 | 34.1 | 1 |
| Ammonium | 10 | 1.769 | 1.00 | 0.254 | 34.1 | 1 |
| Ammonium | 20 | 1.769 | 1.00 | 0.300 | 34.1 | 1 |
| Ammonium | 30 | 1.769 | 1.00 | 0.350 | 34.1 | 1 |
| Ammonium | 40 | 1.430 | 1.17 | 0.376 | 39.8 | 1 |
| Ammonium | 50 | 1.390 | 1.22 | 0.403 | 41.3 | 1 |
| Ammonium | 60 | 1.349 | 1.28 | 0.460 | 43.2 | 1 |
| Ammonium | 70 | 1.302 | 1.36 | 0.520 | 45.7 | 1 |
| Ammonium | 80 | 1.245 | 1.49 | 0.583 | 49.2 | 1 |
| Ammonium | 85 | 1.210 | 1.58 | 0.650 | 52.6 | 1 |
| Ammonium | 90 | 1.165 | 1.73 | 0.794 | 57.6 | 1 |
| Ammonium | 95 | 1.101 | 2.09 | 0.952 | 67.9 | 1 |





## Appendix C    Monthly mean profiles

In order to illustrate the shapes of the actual vertical $\beta^*(z)$ profiles from the model (runs with, ASM, and without assimilation, CTR) and the ceilometers, the 48 individual monthly average profiles are

given in Figures A2, A3, A4 and A5.

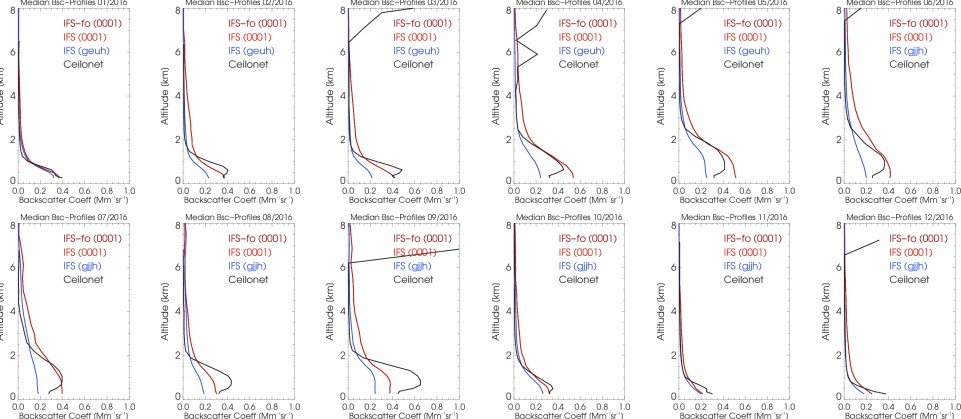

**Fig. A2.** Monthly median profiles 2016 from ceilometer (black), osuite (red) and control run (blue). The median profile from the IFS forward operator is added as a dashed red line.

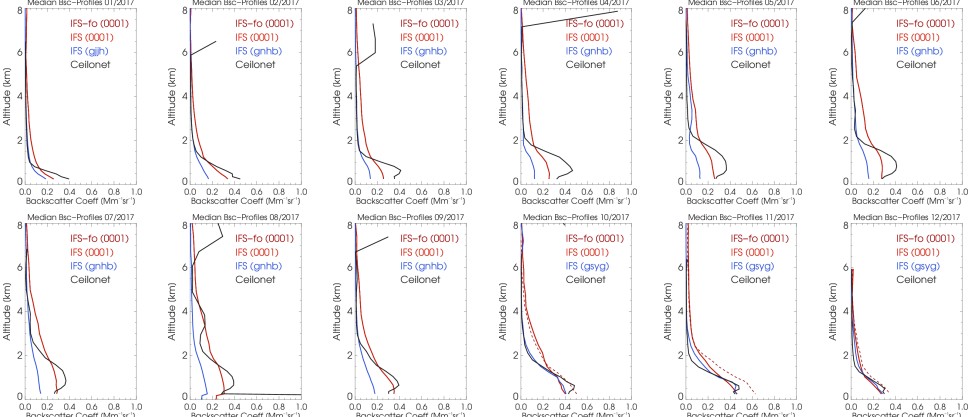

**Fig. A3.** Monthly median profiles 2017 from ceilometer (black), osuite (red) and control run (blue). The median profile from the IFS forward operator is added as a dashed red line.





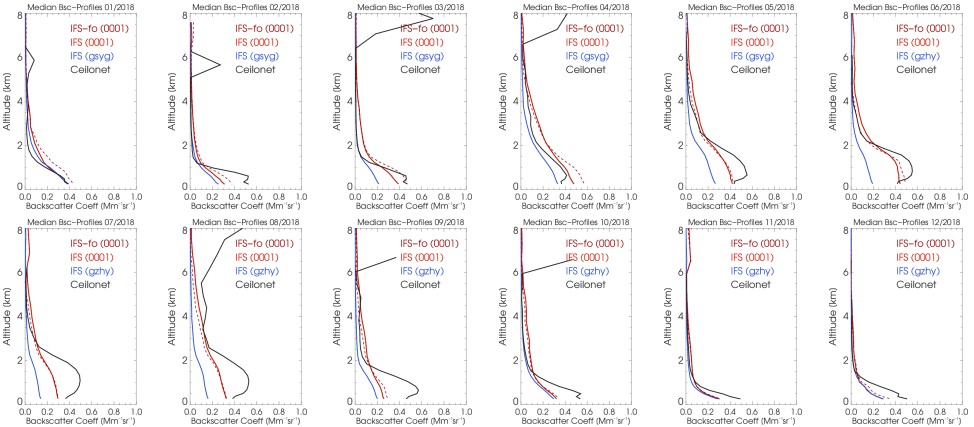

**Fig. A4.** Monthly median profiles 2018 from ceilometer (black), osuite (red) and control run (blue). The median profile from the IFS forward operator is added as a dashed red line.

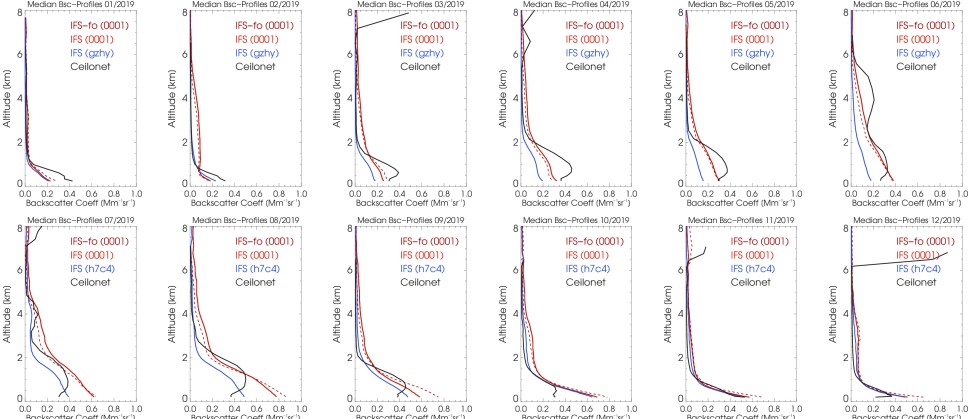

**Fig. A5.** Monthly median profiles 2019 from ceilometer (black), osuite (red) and control run (blue). The median profile from the IFS forward operator is added as a dashed red line.