# Peer review of "Evaluation of ECMWF IFS-AER (CAMS) operational forecasts during cycle 41r1 - 46r1 with calibrated ceilometer profiles over Germany"

_Geoscientific Model Development, 2020_

## Referee Comment (RC1) · Anonymous Referee #1 · 3 Dec 2020

General comments

This paper is on the evaluation of the prediction of aerosols by the air-quality AQ model of the European for Medium-range Weather Forecast (ECMWF) using the ceilometers of the German operational network (CHM15) and observation of the chemical composition of the air done at the station of Hohenpeissenberg (HPB in short). The paper presents the various sources of data, explains the metrics used for the evaluation, and tries to explain the probable origin of the discrepancies observed between the various data sources. This is an interesting paper that points to defaults in the model and give paths to improvements. The observation data span a long, 4-year period, so the statistics are climatologically relevant. The paper contains many results. Its reading is not easy because of the wealth of results on one hand, but also because acronyms sometime appear that have not been introduced (see for instance OM in line 21 on page 1) and several figures are definitely too small for an easy reading (see below). With a few corrections, the reading should become easier. I think the paper deserves publication after minor corrections because of its great interest for AQ model developpers.

Minor comments: Page 1, line 1: OM has not been introduced before. Page 3, line 58: same for BB. Page 7, line 210: according to the Rayleigh scattering theory, the ratio alpha_m/beta_m is strictly equal to 8\pi/3 (not approximately equal). Page 7, last line: the equation is incomplete. Page 9, figure 1: the legend should say specifically that pink dots are stations with 0% of data. Page 11, lines 305-308: the sentence is not clear. Page 11, line 333: the meaning of O(z,t_i) is not explained. Page 14, figure 2: The label on the y-axis of the bottom figures is wrong, it should be % Page 23, figure 5: I think the unit on the y-axis of the middle graph is wrong (should be no unit). Page 25, line 555: the meaning of HPB is given in line 560. Page 27, figure 7: the legend on the graphs is too small. Page 27, line 579: Should be HPB instead of HPS Page 30, figure 8: the graphs are too small. Page 39, line 841: Okt -> Oct.

---

## Referee Comment (RC2) · Anonymous Referee #2 · 6 Dec 2020

The paper deals with comparisons of ceilometer aerosol profile data and aerosol modeling products. This is an important topic. The paper also emphasizes that there is a rather large and well organized ceilometer network in Germany (and Europe) and there are aerosol products that can be used and should be used to further improve modelling and forecast.

In agreement with reviewer #1 (I will not repeat all his/her statements), to my opinion the paper is too long, contains too much information, it is not easy to read, and as a consequence only the absolute experts may read it. And that would be a pity.

It makes no sence to go into much detail. A better separation of main messages (in

the main text part) and information in the Appendix part is necessary. Before that, we need to ask: Do we need all the information, figures, and tables?

A good example for the main point of criticism is the Introduction. The paper is not a review article! So, we want to know already after 2 paragraphs: What are the gaps? What is the problem, we want to work on in this paper? What is the goal of the paper? How is the paper organized, and all this within 1.5-2 pages.

The authors may think about another section, after the introduction, with broader discussion, if necessary.

Some specific points.

You mention cloud formation and wrong forecast....on page 3, please be more specific... liquid water clouds?, ice clouds?, even precipitation....?

On page 4, you mention GALION... what is this?...the same as EARLINET? EARLINET is probably known and should be mentioned.

Sect. 2.1.1, I would skip this subsection, too long, keep it as short as possible. The reader has simly to accept what the retrieval products are.

Page 9, Fig.1, Jenoptik or Lufft...?

Again, keep section2 as a whole as short as possible, move parts into the Appendix... if necessary

Page 14, Fig 2 y-axis, attenuated backscatter, shouldn't that be Mm-1 sr-1?

Page 16, Fig.3: Everything is fine with the ceilometer data down to the ground here? No overlap-related bias, nothing? There is too much information in this figure. Is that all needed? Explanations are so small on printouts.

Section 3.2 up to Section 3.4.3, so many different topics... ! Do we need to present them all?

Section 3 has more than 20 pages ( and the used font is rather small), it is too much!

All in all, the data sets are carefully analyzed. But the list of discussed topics is simply too long.

———————————————

---

## Author Comment (AC1) · 27 Jan 2021

(You may also use the formatted version of answers to your comments in the attached pdf document)

Response to reviewer #1 of the manuscript

'Evaluation of ECMWF IFS-AER (CAMS) operational forecasts during cycle 41r1 - 46r1 with calibrated ceilometer profiles over Germany'

for publication in GMD (MS No.: gmd-2020-308):

We thank the reviewer for her/his efforts to carefully read our manuscript and for the

helpful and constructive comments!

General comments: Following the reviewer's recommendations, we shortened the manuscript main body quite a bit from ∼43 to ∼30 pages, removed redundancies and tried to keep close to the main points. We are confident, that this makes the article easier to read now.

In this we made more use of the appendix, which is now about 9 pages. It disburdens the main text while keeping the information we consider necessary, original and relevant. Particularly section 2.1.1 (description of the mass-to-backscatter calculation) is now in the appendix.

In order to tidy up and ease readability, we removed 4 figures: deleted 2 tiny-looking and dispensable figures from the main text, shifted two figures to the appendix and removed two other figures from the appendix.

We removed (former) Table 2 as it was redundant information with Figure 4, and removed part 2 of Table 3 as this information is provided as numbers in Figure 5.

We increased the font sizes and annotations of all figures and corrected erroneous axistitles.

Specific comments: As you will see in the text, we corrected and followed all issues you raised (properly introduce the acronyms, correct typos and annotations, and remove the tiny-appearing poststamp figures which we ourselves are familiar to use as overview but are admittedly hard to read and identify in detail and interpret for readers outside the lidar community)

Please also note the supplement to this comment:
https://gmd.copernicus.org/preprints/gmd-2020-308/gmd-2020-308-AC1-supplement.pdf
* * *
[Figure]

2020.

---

## Author Comment (AC2) · 27 Jan 2021

(You may also use the formatted version of answers to your comments in the attached pdf document)

Response to reviewer #2 of the manuscript

'Evaluation of ECMWF IFS-AER (CAMS) operational forecasts during cycle 41r1 - 46r1 with calibrated ceilometer profiles over Germany'

for publication in GMD (MS No.: gmd-2020-308):

We thank the reviewer for her/his willingness to carefully read our manuscript and for

the helpful and constructive comments!

General comments:

Following the reviewers recommendations, we shortened the manuscript main body quite a bit from ∼43 to ∼30 pages (now Introduction: 2.5 p, Methodology: 6 p, Results: 13.5 p, Discussion: 7 p, Summary 1 p, References: 7 p, Appendix 9 p). We removed redundancies and tried to keep close to the main points. We are confident, that this makes the article more attractive also to non-lidar-experts and easier readable now.

We are aware of only very few evaluations of the CAMS model with aerosol profile data and receive many questions to this topic at every open discussion in the community. Thus we think that what we now have in section 2 are central elements necessary to understand and classify the results we show. We carefully worked on minimizing it and think we reached a reasonable balance between complication and sophistication.

We made more use of the appendix section, though, which is now about 9 pages. It disburdens the main text while keeping the information we consider necessary, original and relevant.

Following the recommendations we removed 4 figures: deleted 2 tiny-looking and dispensable figures from the main text, shifted two figures to the appendix and removed two other figures from the appendix.

We removed (former) Table 2 as it was redundant information with Figure 4, and removed part 2 of Table 3 as this information is provided as numbers in Figure 5. We increased the fonts and annotations of all figures and corrected typos and erroneous axis-titles.

Specific point-by-point:

- Cloud formation: It is beyond the scope of this article to go into details of this process, although we are aware that it is a bit unsatisfactory to mention it without really digging into it. Thus we specify the term 'water' cloud formation only for the discussion of the

described specific event as this is what we regularly observe at Hohenpeißenberg – formation of stratus clouds in the dust layer near the top of the PBL where ice formation at ambient temperatures is mostly unlikely. According to literature, ice or water nucleation may occur – that's why we do not specify it where we speak about this process in general. We added it here because SD activation gives rise to a marked bias anomaly, deviating strongly from what we usually observe for IFS-AER during events with non-activated Saharan dust. We have no definite observation of rain suppression yet. Unfortunately we can't find a really appropriate reference to this (quite novel) topic.

- We tried to disentangle the information in the main text and the appendix by deleting those appendix Figures A1a and A1b which were largely redundant with Figure 3. The we moved the complete cloud formation case study to the appendix as it is quite independent. In the text we refer only briefly to it's most relevant results and implications.

- GALION (Global Aerosol Lidar Observation Network) and (yes!) EARLINET is now mentioned and explained in the revised version

- Section 2.1.1 (mass-to backscatter conversion) is shifted to the appendix. We consider this section important since the forward operator, including observed and approximated meteorological and physical quantities, is a significant source of uncertainty in our analysis to which we refer in the errors discussion.

- Oh yes surely. . ., Lufft

- See answer to comment on section 2.1.1

- You are right – this was changed

- Yes, the realism of near-ground structure in the observations is an important point: We state repeatedly in the text that we cannot interpret model-observation differences below 300 m a.g. At this height the overlap correction usually is around 50-70% for CHM15k (about factor 2 to 3), which can still reasonably be corrected. In this article the lowest altitude we discuss is thus 400 m above ground. The monthly mean profiles

in Fig 3 are averaged over 21 stations and a month such that all random errors of the instrument-specific overlap functions should cancel out. The consistent tendencies found between 400 and 1000 m a.g. are thus considered quite reliable.

- Figure 3 is one of our two central figures which we think is best suited to illustrate the vertical profile bias in the model. It gives a complementary perspective to Fig 2 and we can't really see how this information could be displayed in a more intuitive and condensed way. We decided to add error bars here to the less relevant control run bias profiles because this is the figure where we think the reader gets the best and easiest feeling for the inherent uncertainties.

- In order to tidy up we removed the fire case from the results section and only briefly mention it in the discussion as it is related to the Saharan dust case on 16/17 Oct 2017 and behaves in IFS-AER in a typical way and can thus be easily described as a quite common case.

- The cloud formation case including figures is moved to the appendix. We unburdened the results section through reducing by ∼40% while still keeping the most relevant information.

- We thought of removing, shifting-to-appendix or even stronger reducing the mixing layer height discussion, but decided to keep it because of the repeatedly experienced large interest for this topic from the scientific (also non-aerosol) community. Even the fact that operational algorithms at present do not provide reliable results for most of the data is confirmed to be relevant in the discussions we have. Thus we think it a useful and keepworthy result that the mixing layer height from the NWP output is remarkably consistent with a manual analysis.

Please also note the supplement to this comment:
https://gmd.copernicus.org/preprints/gmd-2020-308/gmd-2020-308-AC2-supplement.pdf